# Off-Policy Evaluation for Missingness-Aware Policies in MDPs with Rewards Missing Not at Random

**Ziheng Wei** [1]  **Annie Qu** [2]  **Rui Miao** [3]

## Abstract

In offline Reinforcement Learning, immediate rewards in logged batch data are often unobserved due to sparse or irregular record-keeping, or censored beyond certain reward values. This issue arises in practical settings, including health care and marketing. We investigate off-policy evaluation (OPE) in finite-horizon Markov decision processes when rewards are missing not at random (MNAR), which breaks ignorability and induces selection bias even after conditioning on states and actions. To address this, we formalize a reward-dependent propensity model and use future states as shadow variables to identify the full-data conditional mean reward. We further introduce a bridge function that recovers the conditional mean reward without explicitly modeling the MNAR mechanism, and estimate it via a min-max procedure to avoid double sampling. Building upon these identification results, we propose an Fitted-Q-Evaluation-style estimator that propagates the recovered rewards while allowing target policies to depend on past missingness indicators. Finally, we establish consistency and finite-sample error bounds for our OPE estimator, and show through experiments the strong performance of our method compared to existing methods on simulated and MIMIC-III Sepsis data.

## 1. Introduction

Reinforcement Learning (RL) has achieved remarkable successes in sequential decision-making domains ranging from robotics to healthcare, and most recently in large language models and AI agents (Ouyang et al., 2022; Rafailov et al., 2023; Achiam et al., 2023). However, learning optimal policies often requires vast amounts of interaction with the environment, which can be costly, risky, and even unethical in high-stakes real-world applications such as healthcare and education (Tsiatis et al., 2019; Murphy, 2003; Mandel et al., 2014). In these applications, we often have interaction data collected according to some behavior policy, e.g., standard care or usual strategy. Estimating the value of a target policy using historical datasets collected under a different behavior policy, a problem known as Off-Policy Evaluation (OPE), is essential in offline RL (Levine et al., 2020; Uehara et al., 2022; Voloshin et al., 2019; Wang et al., 2024). Accurate OPE is critical for deploying safe and effective policies without the need for dangerous online exploration.

However, in many practical scenarios, such as medical treatment or digital advertising, the data is plagued by unobserved factors or missing values (Little & Rubin, 2019; Kallus & Zhou, 2020). A particularly pervasive challenge arises when rewards are missing not at random (MNAR), under which the probability of observing a reward depends on the latent value of the reward itself. For instance, in health care with scheduled chronic-disease management, patient outcomes such as patient-reported quality-of-life scores may be missing due to incomplete questionnaires or skipped follow-ups, while subsequent clinical states (lab results, vital signs, encounter records) are still recorded in the electronic health record (EHR). The recording of the outcome may itself depend on its value: patients with worsening symptoms are more likely to skip self-reported assessments, while those feeling well may also under-report. Similarly, in multi-touch digital marketing, attribution is frequently disrupted by privacy limitations. While non-conversions (zero rewards) are trivially observable, high-value purchases often involve cross-device journeys, such as a user clicking on mobile but converting on a desktop, or trigger manual review flows that break attribution links. Consequently, high-value conversions go missing while low-value or null outcomes remain fully observed, creating a dataset that systematically biases the learning process against the most desirable outcomes. Similar MNAR feedback has been systematically studied in recommender systems, where popularity and exposure biases make logged interactions MNAR and

[1]Department of Statistics, University of Michigan at Ann Arbor [2]Department of Statistics and Applied Probability, University of California at Santa Barbara [3]Department of Mathematical Sciences, University of Texas at Dallas. Correspondence to: Rui Miao <rui.miao@utdallas.edu>.

*Proceedings of the 43rd International Conference on Machine Learning*, Seoul, South Korea. PMLR 306, 2026. Copyright 2026 by the author(s).

consequently distort offline evaluations (Yang et al., 2018).

Standard OPE methods, such as Fitted Q-evaluation (FQE) and Importance Sampling (IS), rely on fully observed trajectories. Failing to account for the missingness could lead to biased OPE and hence sub-optimal decision-making. In the OPE literature, scheme of missingness has been studied in various aspects. Partially Observable Markov Decision Processes (POMDPs, Kaelbling et al., 1998; Jaakkola et al., 1995) consider the case where the state is partially observed, which can be viewed as a special case of missingness. However, most existing POMDP methods assume that certain state variables are totally unobserved. In general, off-policy value in POMDPs are often unidentifiable without strong assumptions (Tennenholtz et al., 2020; Bennett et al., 2021; Shi et al., 2022; Uehara et al., 2023). Some works define the missingness of certain status (e.g., hitting wall in Gridworld environment (Sutton et al., 1998)) in the reward function by assigning it to be some large negative values to discourage certain actions, which lead to certain states with voided rewards (Ng et al., 1999; Devlin & Kudenko, 2012). However, this approach may not accurately reflect the true reward structure and can introduce bias. Chu et al. (2023); Park et al. (2025) study the OPE problem with truncated trajectories, where they treat missingness as certain constraints. However, by penalizing on the missingness, these methods may shift the policy evaluation away from the true potential reward without missingness. Wang et al. (2025) propose an inverse probability weighting method for OPE with non-ignorable truncation, but their method relies on an extra shadow variable, or requires expert knowledge to select such a variable from observed states.

In this paper, we study the OPE problem in MDPs with MNAR rewards to estimate values of target policies accounting for the past missingness. MNAR rewards break standard ignorability assumptions as the reward-dependent missingness induces selection bias and confounds state-action returns. The challenge is to recover the value of a target policy when the observed trajectories systematically underreport high or low rewards and when the missingness itself can depend on the past action and state, all without online data to re-collect or intervene.

We address these issues by formalizing the reward MNAR mechanism via a reward-dependent propensity score model and leveraging future states as shadow variables. Under mild completeness conditions, the shadow variables allow us to identify the full-data conditional mean reward even when the reward is MNAR. In addition, we introduce a bridge function $b_t(S_t, A_t, S_{t+1})$ satisfying $\mathbb{E}\{b_t(S_t, A_t, S_{t+1}) \mid R_t, S_t, A_t\} = R_t$, enabling recovery of the conditional mean reward without explicitly estimating the MNAR mechanism. This avoids the variance blow-up in inverse propensity weighting. We propose the min-max

optimization to estimate the bridge function and the value function, which avoids the double sampling issue.

Building on these identification results, we further develop an FQE-style estimator that integrates the bridge function and allows target policies to depend on the previous missingness indicators. The procedure propagates the recovered rewards through the Bellman recursion of the target policy, yielding stable value estimates. We further establish the consistency and finite-sample error bounds of the proposed estimator in nonparametric settings. Extensive experiments on simulated data and a MIMIC-III dataset demonstrate the effectiveness of our method compared to existing benchmarks.

## 2. Preliminaries

We consider an episodic Markov Decision Process (MDP) $\mathcal{M} = \{\mathcal{S}, \mathcal{A}, \mathcal{P}, r, T\}$, where $\mathcal{S}$ and $\mathcal{A}$ denote the state and action spaces, respectively. The horizon length $T$ is finite, and we assume the terminal state $S_{T+1}$ is observed. The transition kernels $\mathcal{P} = \{P_t\}_{t=1}^T$ govern the state dynamics, where $P_t : \mathcal{S} \times \mathcal{A} \rightarrow \Delta(\mathcal{S})$ maps state-action pairs to distributions over next states. The reward functions $r = \{r_t\}_{t=1}^T$ are defined as conditional expectations given the next state: $r_t(s, a, s') = \mathbb{E}[R_t \mid S_t = s, A_t = a, S_{t+1} = s']$ for any $(s, a, s') \in \mathcal{S} \times \mathcal{A} \times \mathcal{S}$. We assume bounded rewards $R_t \in \mathcal{R} \subseteq [-1, 1]$.

We introduce an observation indicator $O_t \in \{0, 1\}$, where $O_t = 1$ indicates that the reward $R_t$ is observed at time $t$, and $O_t = 0$ otherwise. Importantly, we allow the rewards to be missing not at random (MNAR), that is, even after conditioning on current states and actions, the missingness probability may depend on the possibly unobserved reward itself. We formalize this through the propensity score $e_t(s, a, r) = P(O_t = 1 \mid S_t = s, A_t = a, R_t = r)$ for $t = 1, \ldots, T$.

**Assumption 2.1** (No Future Dependence). For all $t = 1, \ldots, T - 1$,

$$O_t \perp (S_{t+1:T}, R_{t+1:T}) \mid S_t, A_t, R_t.$$

This assumption states that the current missingness indicator $O_t$, given the current state, action, and reward, is independent of all future states and rewards. For example, in healthcare, whether a patient's health outcome is recorded typically depends on their current condition, not on future events that have not yet occurred.

Our goal is to evaluate the performance of a target policy $\pi = \{\pi_t\}_{t=1}^T$. We allow $\pi_t$ to depend on the previous reward missingness, which is practically relevant when decisions adapt based on whether prior outcomes were observed. For example, in healthcare, a clinician may choose

a more conservative treatment when the previous lab result is missing, whereas the historical standard-care behavior policy acts only on the current clinical state. The two policies thus share the same underlying causal structure but differ in which variables they condition on at decision time. Formally, $\pi_t : \mathcal{S} \times \{0,1\} \to \Delta(\mathcal{A})$ with $\pi_t(a \mid s, o_-) = P(A_t = a \mid S_t = s, O_{t-1} = o_-)$. To accommodate this dependence, we define an augmented state $\widetilde{S}_t = (S_t, O_{t-1}) \in \widetilde{\mathcal{S}} = \mathcal{S} \times \{0,1\}$, so that the target policy can be written as $\pi_t(a \mid \widetilde{S}_t)$. The augmented process must satisfy the Markov property below.

**Assumption 2.2** (Markov Property for Augmented Process). The augmented process $\{(\widetilde{S}_t, A_t)\}_{t=1}^{T}$ with $\widetilde{S}_t = (S_t, O_{t-1})$ is an MDP

$$P(\widetilde{S}_{t+1} \mid \widetilde{S}_{1:t}, A_{1:t}) = P(\widetilde{S}_{t+1} \mid \widetilde{S}_t, A_t), \quad t = 1, \ldots, T.$$

This assumption ensures that augmenting the state with the previous missingness indicator preserves the Markov property, and allows the value function recursion to hold with augmented state. The augmented transition kernel is $P(\widetilde{S}_{t+1} \mid \widetilde{S}_t, A_t) = P((S_{t+1}, O_t) \mid S_t, A_t)$. We set $O_0 = 0$ and let $S_1 \sim \rho_1$ denote the initial state distribution. The Q-function and value function satisfy the Bellman equation

$$\begin{aligned} Q_t^\pi(s,a) =& \mathbb{E}\big[r_t(s, a, S_{t+1}) + V_{t+1}^\pi(S_{t+1}, O_t) \mid S_t = s, \\ & A_t = a\big], \\ V_t^\pi(s, o_-) =& \sum_a \pi_t(a \mid s, o_-) Q_t^\pi(s,a), \quad V_{T+1}^\pi \equiv 0. \end{aligned} \tag{1}$$

The policy value is defined as $V(\pi) \equiv \mathbb{E}_{\widetilde{S}_1 \sim \tilde{\rho}_1}\big[V_1^\pi(\widetilde{S}_1)\big] = E_{S_1 \sim \rho_1}\big[V_1^\pi(S_1, 0)\big]$.

In OPE, data are collected under a behavior policy $\pi^b = \{\pi_t^b\}_{t=1}^{T}$, where $\pi_t^b : \mathcal{S} \to \Delta(\mathcal{A})$ does not depend on the missingness indicators. The observed dataset $\mathcal{D}$ consists of $n$ i.i.d. trajectories $\tau_i = \{S_{t,i}, A_{t,i}, O_{t,i}, R_{t,i}^{\text{obs}}, S_{t+1,i}\}_{t=1}^{T}$ for $i = 1, \ldots, n$, where $R_{t,i}^{\text{obs}} = O_{t,i} \cdot R_{t,i}$ denotes the observed reward (zero when missing). See Figure 1 for a directed acyclic graph (DAG) illustrating the data-generating process.

To deal with distribution shift in OPE, concentrability coefficients are often introduced (Munos, 2003; 2007; Chen & Jiang, 2019; Le et al., 2019; Duan et al., 2021). We define concentrability coefficient $\kappa_t$ at time $t$ in the following assumption.

**Assumption 2.3** (Concentrability). Given target policy $\pi$ and behavior policy $\pi^b$, for each $t = 1, \ldots, T$, assume there exist finite constants $\{\kappa_t\}_{t=1}^{T}$ such that

$$\big\| \frac{\pi_t(a \mid s, o_-)}{\pi_t^b(a \mid s)} \big\|_\infty \le \kappa_t.$$

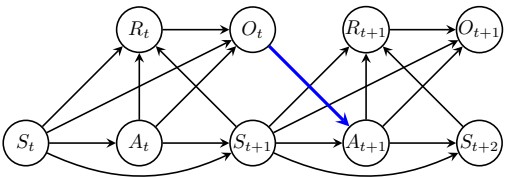

*Figure 1.* DAG for the data-generating process. Black arrows represent the standard MDP dynamics and the MNAR reward mechanism, shared by both policies. The blue arrow $O_t \to A_{t+1}$ is specific to the target policy, which is allowed to depend on the previous missingness indicator at decision time; the behavior policy depends only on the current state.

Equivalently, for all $(s, o_-, a)$ with $\pi_t^b(a \mid s) > 0$, $\pi_t(a \mid s, o_-) \le \kappa_t \pi_t^b(a \mid s)$.

**Assumption 2.4.** Assume there exist constants $a > 0$ and $\alpha_t \ge \alpha > 1$ such that for all $t \in \{1, \ldots, T\}$,

$$\kappa_t \le 1 + \frac{a}{t^{\alpha_t}}.$$

Assumption 2.4 is an enhanced version for Assumption 2.3, which controls the cumulative growth of the concentrability coefficients so that the action mismatch does not compound over time. A structurally similar decay condition appears in Assumption 2 of Miao et al. (2022) in a POMDP setting.

**Corollary 2.5** (Bounded cumulative concentrability). *Under Assumptions 2.3 and 2.4, for $t = 1, \ldots, T$,*

$$(\prod_{j=1}^{t} \kappa_j)^{1/2} \le \exp\Big(\frac{1}{2} \sum_{j=1}^{t} \frac{a}{j^{\alpha_j}}\Big) \le \exp\big(\frac{a}{2} \zeta(\alpha)\big) := K,$$

*where $\zeta(\cdot)$ is the Riemann zeta function.*

## 3. Identification

In this section, we establish identification results for the policy value, and formalize the conditions required for our approach.

To address the challenges posed by MNAR data in causal inference, Miao et al. (2015); Miao & Tchetgen Tchetgen (2016) propose identification methods with the help of auxiliary variables called *shadow variables*. Inspired by this line of research, we establish a nonparametric value-based approach for policy value identification. Our key insight is to adopt the next state $S_{t+1}$ as the shadow variable, which serves as a proxy that helps recover information about the unobserved rewards. The shadow variable must satisfy two conditions that govern its relationship with the reward and missingness indicator.

**Assumption 3.1** (Exclusion Restriction). Suppose for all $t = 1, \ldots, T$, $S_{t+1}$ satisfies

$$S_{t+1} \perp O_t \mid R_t, S_t, A_t.$$

**Assumption 3.2** (Relevance Condition). Suppose for all $t = 1, \ldots, T$, $S_{t+1}$ satisfies

$$S_{t+1} \not\perp R_t \mid S_t, A_t, O_t = 1.$$

The two assumptions above are basic conditions for $S_{t+1}$ to be a valid shadow variable at time $t$. Assumption 3.1 is a direct consequence of Assumption 2.1, which shows that conditional on the current state-action pair and the (possibly unobserved) reward, $S_{t+1}$ provides no additional information about whether the reward is observed. Assumption 3.2 ensures that on the observed subset, $S_{t+1}$ remains informative about $R_t$ beyond what is already captured by $(S_t, A_t)$. This condition guarantees that the shadow variable carries useful information about the reward. In the causal graph in Figure 1, Assumption 3.1 is consistent with $d$-separation: conditioning on $(R_t, S_t, A_t)$ blocks all paths from $S_{t+1}$ to $O_t$. For subsequent analysis, we define the extended propensity score of non-missingness $e_t(s, a, r, s') = P(O_t = 1 \mid S_t = s, A_t = a, R_t = r, S_{t+1} = s')$. The choice of $S_{t+1}$ aligns with the recurring idea in POMDPs, i.e., leveraging future states or observations to serve as proxy latent state information (Singh et al., 2003; Uehara et al., 2023; Xu et al., 2023; Zhang & Jiang, 2024). Moreover, $S_{t+1}$ is endogenous to the MDP and already recorded in the logged transitions, so it requires no auxiliary measurement beyond the standard offline data, in contrast to approaches that specify an extra shadow variable (Wang et al., 2025).

For identification, rather than explicitly modeling the missingness mechanism under MNAR, our goal is to recover the full-data conditional mean reward $\mathbb{E}[R_t \mid S_t, A_t]$ using only observable quantities. Under MNAR, the observed reward conditional expectation $\mathbb{E}[R_t \mid S_t = s, A_t = a, O_t = 1]$ generally differs from the target $\mathbb{E}[R_t \mid S_t = s, A_t = a]$, and directly using observed rewards would lead to a biased policy evaluation.

We adopt a bridge-based imputation strategy for missing rewards motivated by proximal causal inference (Tchetgen Tchetgen et al., 2020; Cui et al., 2024). Related bridge constructions also appear in the literature on confounded POMDPs (Miao et al., 2022; Shi et al., 2022; Hong et al., 2023; Li et al., 2025). The core idea is to construct functions that can learn the missing rewards in an unbiased manner by exploiting the relationship between rewards and next states.

Specifically, we introduce a sequence of bridge functions $\{b_t : \mathcal{S} \times \mathcal{A} \times \mathcal{S} \to \mathbb{R}\}_{t=1}^T$ satisfying the moment condition that

$$\mathbb{E}[b_t(S_t, A_t, S_{t+1}) \mid R_t, S_t, A_t] = R_t, \quad a.s. \quad (2)$$

Equation (2) links the target quantity of interest to the observable offline distribution, and converts the recovery of $\mathbb{E}[R_t \mid S_t, A_t]$ into an estimable conditional moment problem.

Taking conditional expectation of (2) given $(S_t, A_t)$ yields

$$\mathbb{E}[b_t(s, a, S_{t+1}) \mid S_t = s, A_t = a]$$
$$= \mathbb{E}(R_t \mid S_t = s, A_t = a) := \bar{r}_t(s, a). \quad (3)$$

Thus, the bridge function reproduces the correct one-step conditional mean reward required by the Bellman recursion.

Moreover, a crucial observation which enables practical estimation is that

$$P(S_{t+1} \mid R_t, S_t, A_t, O_t = 1) = P(S_{t+1} \mid R_t, S_t, A_t),$$

by Assumption 2.1. This implies that the bridge moment condition in Equation (2) can be identified from the observed subset $\{O_t = 1\}$. This means we can estimate the bridge function $b_t$ using only samples where rewards are observed, and then evaluate it at samples with $O_t = 0$ to impute the missing rewards.

We further introduce the following assumptions for the identification of the policy value.

**Assumption 3.3** (Positivity). For all $t = 1, \ldots, T$, and for all $(s, a, r) \in \mathcal{S} \times \mathcal{A} \times \mathcal{R}$, $0 < e_t(s, a, r) < 1$.

The positivity assumption ensures that every state-action-reward triple has a positive probability of being both observed and unobserved, which is commonly used in causal inference literature.

**Assumption 3.4** (Completeness). For all $(s, a) \in \mathcal{S} \times \mathcal{A}$, $t = 1, \ldots, T$,

(1) For any square-integrable function $h$,

$$\mathbb{E}[h(R_t) \mid S_t = s, A_t = a, S_{t+1}]$$
$$= \int h(R_t) p(R_t \mid s, a, S_{t+1}) dR_t = 0, \quad a.s.$$

if and only if $h(R_t) = 0, \quad a.s.$;

(2) For any square-integrable function $g$,

$$\mathbb{E}[g(S_{t+1}) \mid R_t, S_t = s, A_t = a]$$
$$= \int g(S_{t+1}) p(S_{t+1} \mid R_t, s, a) dS_{t+1} = 0, \quad a.s.$$

if and only if $g(S_{t+1}) = 0, \quad a.s.$

Assumption 3.4 guarantees the existence and uniqueness of the bridge functions $b_t$, for $t = 1, \ldots, T$. Completeness assumptions are standard in the proximal causal inference (Tchetgen Tchetgen et al., 2020; Cui et al., 2024), where they ensure that the conditional expectations are sufficiently rich to identify the target functional.

To provide concrete intuition, we characterize completeness in the tabular setting.

**Example 3.5** (Completeness in tabular setting). *Assume $\mathcal{S}$, $\mathcal{R}$ are tabular. Let matrix $M_{t,s,a} \in \mathbb{R}^{|\mathcal{S}| \times |\mathcal{R}|}$ where $M_{t,s,a}(s', r) := P(S_{t+1} = s' \mid R_t = r, S_t = s, A_t = a)$, $s' \in \mathcal{S}, r \in \mathcal{R}$. Then*

1. *If for all $(t, s, a)$, $rank(M_{t,s,a}) = |\mathcal{R}|$, then Assumption 3.4 (1) holds, and hence the bridge exists;*

2. *If for all $(t, s, a)$, $rank(M_{t,s,a}) = |\mathcal{S}|$, then Assumption 3.4 (2) holds, and hence the bridge is unique;*

3. *If $|\mathcal{S}| = |\mathcal{R}|$, then $M_{t,s,a}$ is invertible for all $(t, s, a)$ if and only if both Assumption 3.4 (1) and Assumption 3.4 (2) hold.*

Then we give the identification results for policy value as follows:

**Theorem 3.6** (Policy value identification). *For an augmented MDP satisfying Assumptions 2.1, 2.2 and 3.2 to 3.4 and some regularity conditions, there always exist bridges $\{b_t\}_{t=1}^T$ that satisfy Equation (2), and the policy value then can be identified using $\{b_t\}_{t=1}^T$.*

See Appendix D for other regularity conditions and proof.

Based on Theorem 3.6, we develop a value-based approach for policy value identification, which circumvents modeling the missing mechanism explicitly, in contrast to approaches such as Miao & Tchetgen Tchetgen (2016); Miao & Tchetgen (2018); Wang et al. (2025). This is practically significant as it can avoid the high variance induced from IPW methods or requires strong parametric assumptions about the missingness. Our identification procedure consists of three steps.

**Step 1 (Learn $b_t$).** For each $t = 1, \ldots, T$, learn $b_t$ from

$$\mathbb{E}[b_t(s, a, S_{t+1}) \mid R_t = r, S_t = s, A_t = a] = r,$$

using the observed subset with $O_t = 1$. This step leverages the shadow variable structure to extract reward information from state transitions.

**Step 2 (Identify $Q^\pi$ and $V^\pi$).** Define the imputed reward

$$\widetilde{R}_t := O_t R_t + (1 - O_t) b_t(S_t, A_t, S_{t+1}), \tag{4}$$

and solve the Bellman recursion

$$Q_t^\pi(s, a) = \mathbb{E}[\widetilde{R}_t + V_{t+1}^\pi(S_{t+1}, O_t) \mid S_t = s, A_t = a],$$
$$V_t^\pi(s, o_-) = \sum_a \pi_t(a \mid s, o_-) Q_t^\pi(s, a), \quad V_{T+1}^\pi \equiv 0.$$

It is trivial to verify that

$$\mathbb{E}[\widetilde{R}_t \mid R_t, S_t, A_t] = R_t, \quad a.s. \tag{5}$$

**Step 3 (Identify the policy value).** Compute $V(\pi) = \mathbb{E}_{\widetilde{S}_1 \sim \widetilde{\rho}_1}[V_1^\pi(\widetilde{S}_1)]$ by backward induction.

## 4. Estimation

In this section, we discuss estimation of policy value and propose a FQE-style estimation method. To estimate the policy value, it suffices to estimate the bridge functions $\{b_t\}$ from conditional moment models

$$\mathbb{E}[b_t(S_t, A_t, S_{t+1}) - R_t \mid R_t, S_t, A_t, O_t = 1] = 0, \quad a.s., \tag{6}$$

which can be viewed as nonparametric instrumental variable (NPIV) problems. A natural approach would be to directly minimize the squared conditional moment

$$\min_{b_t \in \mathcal{B}^{(t)}} \mathbb{E}\left[\left(\mathbb{E}[b_t(S_t, A_t, S_{t+1}) - R_t \mid R_t, S_t, A_t, O_t = 1]\right)^2\right].$$

However, this is not implementable for a single batch of trajectories because the squared conditional moments can lead to the double-sampling issue (Baird et al., 1995; Sutton et al., 1998). To circumvent the double-sampling problem, we adopt a min-max estimator for $b_t$ (Dikkala et al., 2020). The key insight is to replace the squared conditional moment with a saddle-point formulation that can be estimated from a single batch of samples.

For each time step $t$, we solve

$$\min_{b_t \in \mathcal{B}^{(t)}} \sup_{g \in \mathcal{G}^{(t)}} \frac{1}{n_t} \sum_{i \in \mathcal{I}_t^{\text{obs}}} \left[(b_t(S_{t,i}, A_{t,i}, S_{t+1,i}) - R_{t,i}) g_t(R_{t,i}, S_{t,i}, A_{t,i})\right] - \lambda(\|g_t\|_{\mathcal{G}^{(t)}}^2 + \frac{U}{\delta^2}\|g_t\|_2^2) + \lambda\mu\|b_t\|_{\mathcal{B}^{(t)}}^2, \tag{7}$$

where $\mathcal{I}_t^{\text{obs}} = \{i \in \{1, \ldots, n\} : O_{t,i} = 1\}$ denotes the observed dataset at time $t$, and $n_t = |\mathcal{I}_t^{\text{obs}}|$. We denote the function classes of $g_t$ and $b_t$ by $\mathcal{G}^{(t)}$, $\mathcal{B}^{(t)}$, which can be chosen as finite dimensional linear spaces, and infinite dimensional spaces like RKHSs, neural networks, etc. We focus on RKHSs in this paper. Let $\mathcal{Q}^{(t)}$ be the RKHS containing function $Q_t$. The term $\lambda \frac{U}{\delta^2}\|g_t\|_2^2$ is the $L_2$ penalty on the critic function $g_t$. The norms $\|\cdot\|_{\mathcal{G}^{(t)}}^2$, $\|\cdot\|_{\mathcal{B}^{(t)}}^2$, $\|\cdot\|_{\mathcal{Q}^{(t)}}^2$ denote the functional norm associated with $\mathcal{G}^{(t)}$, $\mathcal{B}^{(t)}$, $\mathcal{Q}^{(t)}$. $\lambda, U, \delta, \mu > 0$ are tuning parameters for the penalties.

Then, we can substitute the estimates into fitted-Q-evaluation (FQE) algorithm and obtain the estimate of policy value $\widehat{V}(\pi)$. See Algorithm 1 for the point-estimated policy value estimation algorithm, where we use penalized nonparametric least squares to learn $Q_t$:

$$\widehat{Q}_t = \arg\min_{f \in \mathcal{Q}^{(t)}} \frac{1}{n} \sum_{i=1}^n \left(f(S_{t,i}, A_{t,i}) - y_{t,i}\right)^2 + \lambda_{Q,t}\|f\|_{\mathcal{Q}^{(t)}}^2, \tag{8}$$

where $y_{t,i}$ is defined in Algorithm 1.

**Algorithm 1** Proximal FQE algorithm

**Input:** Offline dataset $\mathcal{D} = \{\tau_i\}_{i=1}^n$, where $\tau_i = \{(S_{t,i}, A_{t,i}, O_{t,i}, R_{t,i}^{\mathrm{obs}}, S_{t+1,i})\}_{t=1}^T$, target policy $\pi = \{\pi_t\}_{t=1}^T$, horizon $T$, function classes $\{\mathcal{B}^{(t)}, \mathcal{G}^{(t)}, \mathcal{Q}^{(t)}\}$.
**Initialize:** $\widehat{V}_{T+1}^\pi(\cdot, \cdot) \leftarrow 0$.
**for** $t = T$ down to $1$ **do**
  **Bridge fitting:** Obtain $\hat{b}_t$ by solving Equation (7) on $\mathcal{I}_t^{\mathrm{obs}}$.
  **Imputation:** for all $i = 1, \ldots, n$ set $\widehat{\widetilde{R}}_{t,i} \leftarrow R_{t,i}^{\mathrm{obs}} + (1 - O_{t,i})\hat{b}_t(S_{t,i}, A_{t,i}, S_{t+1,i})$.
  **Targets for Bellman regression:**
  **if** $t < T$ **then**
    $y_{t,i} \leftarrow \widehat{\widetilde{R}}_{t,i} + \widehat{V}_{t+1}^\pi(S_{t+1,i}, O_{t,i}), i = 1, \ldots, n.$
  **else**
    $y_{t,i} \leftarrow \widehat{\widetilde{R}}_{t,i}, i = 1, \ldots, n.$
  **end if**
  **Fit $Q_t$:** regress $y_{t,i}$ on $(S_{t,i}, A_{t,i})$ by Equation (8) to obtain $\widehat{Q}_t$.
  **Define $V_t^\pi$:** $\widehat{V}_t^\pi(s, o_-) \leftarrow \sum_a \pi_t(a \mid s, o_-)\widehat{Q}_t(s, a)$.
**end for**
**Output:** $\widehat{V}(\pi) \leftarrow \frac{1}{n} \sum_{i=1}^n \widehat{V}_1^\pi(S_{1,i}, 0)$.

# 5. Theoretical results

In this section, we establish consistency and finite-sample estimation error bounds for bridges $\hat{b}_t$ and the policy value.

## 5.1. Preliminaries

**Definition 5.1** (Local Rademacher Complexity (Bartlett et al., 2005)). *For any function class $\mathcal{G}$ defined over random variable $X$ and radius $\delta > 0$, the local Rademacher complexity is*

$$\mathcal{R}_n(\mathcal{G}, \delta) = \mathbb{E}_{\varepsilon, X}\left[\sup_{g \in \mathcal{G}: \|g\|_2 \le \delta} \left|\frac{1}{n}\sum_{i=1}^n \varepsilon_i g(X_i)\right|\right],$$

where $\{X_i\}$ are i.i.d. samples of $X$ and $\{\varepsilon_i\}$ are Rademacher random variables. $\|g\|_2^2 := \mathbb{E}[g(X)^2]$ is the $L_2$ norm of function $g$.

Suppose the function class $\mathcal{G}$ satisfies

1. *symmetric*, if $g \in \mathcal{G}$ then $-g \in \mathcal{G}$;

2. *star-shaped*, if $g \in \mathcal{G}$ then $rg \in \mathcal{G}$ for all $r \in [0, 1]$;

3. *b-uniformly bounded*, $\|g\|_\infty := \sup_{x \in \mathcal{X}} |g(x)| \le b$ for all $g \in \mathcal{G}$.

Then, the critical radius of such function class $\mathcal{G}$, denoted by $\delta_n$, is the smallest solution to the inequality $\mathcal{R}_n(\mathcal{G}, \delta) \le \frac{\delta^2}{b}$.

## 5.2. Bridge function estimation error bound

For notational simplicity, define the projection operator $\mathcal{T}_t : \mathcal{L}^2\{\mathcal{S} \times \mathcal{A} \times \mathcal{S}\} \to \mathcal{L}^2\{\mathcal{R} \times \mathcal{S} \times \mathcal{A}\}$, which satisfies

$$\mathcal{T}_t b_t = \mathbb{E}[b_t(S_t, A_t, S_{t+1})|R_t, S_t, A_t].$$

**Assumption 5.2** (Boundedness of $\mathcal{T}_t$). For any $b_t \in \mathcal{B}^{(t)}$, $\mathcal{T}_t b_t \in \mathcal{G}^{(t)}$, and there exists $L > 0$ such that

$$\|\mathcal{T}_t b_t\|_{\mathcal{G}^{(t)}} \le L\|b_t\|_{\mathcal{B}^{(t)}}.$$

**Assumption 5.3** (Realizability). Suppose the true bridge function $b_t^*$ lies in function class $\mathcal{B}^{(t)}$. Similarly, we also assume $Q_t^\pi \in \mathcal{Q}^{(t)}$.

In practice, the boundedness assumption often holds when the conditional distribution of $S_{t+1}$ given $(R_t, S_t, A_t)$ is sufficiently smooth. Also, Realizability is a standard assumption in nonparametric estimation, requiring that the function classes are rich enough to contain the true targets.

Next, we define the $B_t$-bounded norm subset of $\mathcal{B}^{(t)}$ as $\mathcal{B}_B^{(t)} := \{b_t \in \mathcal{B}^{(t)} : \|b_t\|_{\mathcal{B}^{(t)}} \le B_t\}$ and $U_t$-bounded norm subset of $\mathcal{G}^{(t)}$ as $\mathcal{G}_U^{(t)} := \{g_t \in \mathcal{G}^{(t)} : \|g_t\|_{\mathcal{G}^{(t)}} \le U_t\}$.

**Assumption 5.4** (Richness of test function class). We suppose the test function approximation error within subset $\mathcal{G}_{L^2\|(b-b_t^*)\|_{\mathcal{B}^{(t)}}^2}^{(t)}$ is bounded by

$$\sup_{b \in \mathcal{B}_B^{(t)}} \inf_{g_t \in \mathcal{G}_{L^2\|b-b_t^*\|_{\mathcal{B}^{(t)}}^2}^{(t)}} \|g_t - \mathcal{T}_t(b - b_t^*)\|_2 \le \eta_t < \infty.$$

This shows that function class $\mathcal{G}^{(t)}$ is rich enough so that $\mathcal{T}_t(b - b_t^*)$ admits an $L_2$-approximation within $\mathcal{G}_{L^2\|(b-b_t^*)\|_{\mathcal{B}^{(t)}}^2}^{(t)}$ uniformly over $b \in \mathcal{B}_B^{(t)}$.

Now we are ready to analyze min-max estimator $\hat{b}_t$ estimated by Equation (7).

**Theorem 5.5** (Projected bridge estimation error bound (Dikkala et al., 2020; Miao et al., 2022)). *For any $t = 1, \ldots, T$, suppose function class $\mathcal{G}^{(t)}$ is star-shaped and symmetric. Suppose $\mathcal{G}^{(t)}$ and $\mathcal{B}^{(t)}$ are 1-uniformly bounded. Define product class*

$$\mathcal{J}_{B,U}^{(t)} := \Big\{((s, a, s'), (r, s, a)) \mapsto \alpha(b_t(s, a, s') - b_t^*(s, a,$$
$$s'))g_{b,t}^U(r, s, a) \mid b_t - b_t^* \in \mathcal{B}_B^{(t)}, \alpha \in [0, 1]\Big\},$$

*where $g_{b,t}^U = \arg\min_{g_t \in \mathcal{G}_U^{(t)}} \|g_t - \mathcal{T}_t(b - b_t^*)\|_2$. Define the two critical radii for function class $\mathcal{G}_{3U}^{(t)}$ and $\mathcal{J}_{B,L^2B}^{(t)}$, namely $\delta_t^{\mathcal{G}}$ and $\delta_t^{\mathcal{J}}$, and their maximum $\delta_{n_t} := \max\{\delta_t^{\mathcal{G}}, \delta_t^{\mathcal{J}}\}$. Let $\delta_t = \delta_{n_t} + c_0\sqrt{\frac{\log(c_1/\zeta)}{n_t}}$. Assume $\eta_t \lesssim \delta_t$, then if $\lambda \asymp \frac{\delta_t^2}{U}$ and $\mu \ge \frac{4}{3}L^2 + \frac{36}{B_t\lambda}\delta_t^2$, we have with probability $1 - 3\zeta$, the following bound holds*

$$\|\mathcal{T}_t(\hat{b}_t - b_t^*)\|_2 \lesssim \delta_t \max\{1, \|b_t^*\|_{\mathcal{B}^{(t)}}^2\}.$$

**Corollary 5.6** (RKHS cases with polynomial eigen decay). *We further suppose $\mathcal{B}^{(t)}$ and $\mathcal{G}^{(t)}$ are RKHSs for all $t = 1, \ldots, T$. $K_{B,t}$ and $K_{G,t}$ are the kernels of $\mathcal{B}^{(t)}$ and $\mathcal{G}^{(t)}$ with non-increasing eigenvalues $\{\mu_{t,j}^{\mathcal{B}}\}_{j=1}^{\infty}$ and $\{\mu_{t,j}^{\mathcal{G}}\}_{j=1}^{\infty}$. We assume polynomial decay on eigenvalues, i.e., for some $\alpha_B, \alpha_G > \frac{1}{2}$, $\mu_{t,j}^{\mathcal{B}} \lesssim j^{-2\alpha_B}$, $\mu_{t,j}^{\mathcal{G}} \lesssim j^{-2\alpha_G}$, $j \to \infty$. Let $\alpha_{\min} := \min\{\alpha_B, \alpha_G\}$. Then the critical radius in Theorem 5.5 satisfy $\delta_{n_t} \lesssim \max\{\sqrt{U_t}, LB_t\} n_t^{-\frac{\alpha_{\min}}{2\alpha_{\min}+1}} \log n_t$. Consequently, under the conditions of Theorem 5.5, for all $t = 1, \ldots, T$, with probability at least $1 - 3\zeta$,*

$$\|\mathcal{T}_t(\hat{b}_t - b_t^*)\|_2 \lesssim \sqrt{\log(c_1/\zeta)} n_t^{-\frac{\alpha_{\min}}{2\alpha_{\min}+1}} \log n_t.$$

Theorem 5.5 provides a finite-sample bound for the projected error $\|\mathcal{T}_t(\hat{b}_t - b_t^*)\|_2$, where linear operator $\mathcal{T}_t$ maps a bridge function to a conditional expectation given $(R_t, S_t, A_t)$. However, for downstream analysis we also need control of the rooted mean-squared error (RMSE) $\|\hat{b}_t - b_t^*\|_2$.

In general, converting projected error bounds into $L_2$ error bounds is nontrivial because conditional moment problems are typically ill-posed inverse problems: the operator $\mathcal{T}_t$ is often compact and hence may not admit a stable inverse on an unrestricted function class. This phenomenon and the role of regularization in such conditional moment models are well-studied in the semi-/nonparametric literature; see, e.g., (Chen & Reiss, 2011; Chen & Pouzo, 2012). We introduce an ill-posedness measure for the conditional expectation operator $\mathcal{T}_t$, following the definition in Dikkala et al. (2020). Since the true bridge $b_t^*$ is not assumed to lie in $\mathcal{B}_B^{(t)}$, we define the best approximation within the ball

$$b_{t,*} := \arg \min_{b \in \mathcal{B}_B^{(t)}} \|b - b_t^*\|_2, \quad \varepsilon_t(B_t) := \inf_{b \in \mathcal{B}_B^{(t)}} \|b - b_t^*\|_2.$$

**Definition 5.7** (Measure of ill-posedness). Define the ill-posedness coefficient $\tau_t(B_t) := \sup_{b \in \mathcal{B}_B^{(t)}} \frac{\|b - b_{t,*}\|_2}{\|\mathcal{T}_t(b - b_{t,*})\|_2}$, and assume $\tau_t(B_t) < \infty$.

By combining Theorem 5.5 with Definition 5.7, we obtain an $L_2$ error bound for the bridge estimator:

$$\|\hat{b}_t - b_t^*\|_2 \leq \tau_t(B_t)\delta_t + (\tau_t(B_t) + 1)\varepsilon_t(B_t).$$

The choice of $B_t$ trades off the approximation bias $\varepsilon_t(B_t)$ and the ill-posedness factor $\tau_t(B_t)$. Consider the whole function class $\mathcal{B}^{(t)}$, where $b_{t,*} = b_t^*$ and $\varepsilon_t(B_t) = 0$ under Assumption 5.3, this gives the global ill-posedness

$$\tau_t := \sup_{b \in \mathcal{B}^{(t)}} \frac{\|b - b_{t,*}\|_2}{\|\mathcal{T}_t(b - b_{t,*})\|_2},$$

where we assume $\tau_t < \infty$. The RMSE is given by $\|\hat{b}_t - b_t^*\|_2 \lesssim \tau_t \delta_t$.

**Theorem 5.8** (Bridge estimation error bound). *For any $t = 1, \ldots, T$, suppose function class $\mathcal{G}^{(t)}$ is star-shaped and symmetric. Suppose $\mathcal{G}^{(t)}$ and $\mathcal{B}^{(t)}$ are 1-uniformly bounded. Consider min-max estimator $\hat{b}_t$ estimated by Equation (7). Define function classes $\text{star}\left(\mathcal{B}^{(t)} - b_t^*\right) = \{r(b - b_t^*) : b - b_t^* \in \mathcal{B}_B^{(t)}, r \in [0,1]\}$, and $\text{star}\left(\mathcal{T}_t(\mathcal{B}^{(t)} - b_t^*)\right) = \{r g_{b,t}^U : b - b_t^* \in \mathcal{B}_B^{(t)}, r \in [0,1]\}$, where $g_{b,t}^U = \arg \min_{g_t \in \mathcal{G}_U^{(t)}} \|g_t - \mathcal{T}_t(b - b_t^*)\|_2$. Define the $\delta_{n_t}$ as the upper bound on the critical radii of $\mathcal{G}_{3U}^{(t)}$ and the two function classes. Let $\delta_t = \delta_{n_t} + c_0 \sqrt{\frac{\log(c_1/\zeta)}{n_t}}$. Assume $\eta_t \lesssim \delta_t$, then if $\lambda \asymp \frac{\delta_t^2}{U}$ and $\mu \geq \frac{4}{3}L^2 + \frac{36}{B_t\lambda}\delta_t^2$, then with probability $1 - 3\zeta$, the following bound holds*

$$\|\hat{b}_t - b_t^*\|_2 \lesssim \tau_t \delta_t \max\{1, \|b_t^*\|_{\mathcal{B}^{(t)}}^2\}.$$

### 5.3. Policy value estimation error bound

Based on Theorem 5.8, we can further bound the OPE error of the policy value $\widehat{V}(\pi)$ estimated from *Algorithm 1*.

**Theorem 5.9** (Policy value estimation error bound). *Suppose RKHSs $\mathcal{Q}^{(t)}, \mathcal{B}^{(t)}, \mathcal{G}^{(t)}$ have polynomial eigen-decay rate*

$$\mu_{t,j}^{\mathcal{Q}} \lesssim j^{-2\alpha_Q}, \mu_{t,j}^{\mathcal{B}} \lesssim j^{-2\alpha_B}, \mu_{t,j}^{\mathcal{G}} \lesssim j^{-2\alpha_G},$$

*where $\alpha_Q, \alpha_B, \alpha_G > 1/2$. Define $\alpha_{\min} = \min\{\alpha_Q, \alpha_B, \alpha_G\} > \frac{1}{2}$. Denote $\delta_{t,*} = \bar{\delta}_{t,*} + c_0 \sqrt{\frac{\log(c_1 T/\zeta)}{n}}$ for some $c_0, c_1 > 0$ where $\bar{\delta}_{t,*}$ is the upper bound of the critical radii of difference classes $\Delta \mathcal{Q}^{(t)}$, $\Delta \mathcal{Q}^{(t+1)}$, $\Delta \mathcal{B}^{(t)}$ and $\mathcal{G}_U^{(t)}$ defined in Appendix G. Suppose $\lambda_{Q,t} \asymp (\delta_{\Delta \mathcal{Q}}^{(t)})^2$ and let $\tau_{\max} = \max_{t \leq T} \tau_t$. Under Assumptions 2.3, 2.4, 5.2 and 5.3 and assumptions for Theorem 3.6, with probability at least $1 - \zeta$, the policy value estimation error is bounded by*

$$\left|\widehat{V}(\pi) - V(\pi)\right| \lesssim K \tau_{\max} T^2 \sqrt{\log(c_1 T/\zeta)} n^{-\frac{\alpha_{\min}}{2\alpha_{\min}+1}} \log n.$$

Without considering the ill-posedness, our OPE error bound achieves the optimal rate in $n$ in the classical nonparametric regression (Stone, 1982). Our error bound exhibits a $T^2$ dependence on the horizon, which arises from error propagation through the Bellman recursion and the additional complexity introduced by estimating bridge functions under the MNAR setting. For comparison, Wang et al. (2024) provide a fine-grained analysis of FQE under fully observed rewards. Under the completeness assumption for $Q$-functions alone, they establish an error bound of order $\mathcal{O}(T^{1.5}\sqrt{1/n})$ for both parametric and nonparametric settings, improving upon the $\mathcal{O}(T^2\sqrt{\kappa/n})$ bounds in prior work (Duan et al., 2020; Zhang et al., 2022). With an additional realizability assumption on the probability ratio functions $w_t^{\pi}$, the rate further

improves to $\mathcal{O}(T\tilde{\kappa}\sqrt{1/n})$, matching the sharpest known bound under the tabular setting (Yin & Wang, 2020). The additional $T$ factor in our bound compared to the $T^{1.5}$ rate in Wang et al. (2024) partially reflects the cost of correcting for MNAR rewards through the bridge function mechanism.

# 6. Experiments

In this section, we evaluate the performance of the proposed OPE estimator with rewards MNAR on both simulated and real-world data. We compare our method with four baselines: an IPW-based OPE method (Wang et al., 2025), a naive FQE baseline, an imputation-based OPE method and a reward-shaping-based OPE method (Parbhoo et al., 2020). The naive FQE estimator ignores the MNAR mechanism and applies FQE directly to the observed rewards. The IPW baseline adapts the weighting scheme of Wang et al. (2025), which was originally developed for a trajectory dropout model. The imputation-based baseline learns the reward function by regression and imputes the unobserved reward using the fitted model. The reward-shaping baseline, named SCOPE, uses per-step importance sampling with potential-based reward shaping.

We present results on simulation studies and a real-data application below. Our code is available at https://github.com/NAIVlab/ShadOPE.

## 6.1. Simulation studies

We conduct simulation studies in finite-horizon episodic MDPs to evaluate the proposed method under varying missingness levels and reward generation mechanisms. We compare ProxFQE against four baselines, naive FQE, IPW-FQE, Impute-FQE, and SCOPE, over 50 random seeds.

We set state $S_t = (S_{t,1}, S_{t,2})^\top \in \mathcal{S} = \mathbb{R}^2$ as a two-dimensional vector. The action space is binary, $\mathcal{A} = \{-1, 1\}$. Let $\mathcal{O} = \{0, 1\}$ and $\mathcal{R} = \mathbb{R}$. The propensity score is set as

$$e_t(S_t, A_t, R_t) = \text{expit}\left(c_0 - 0.1A_t + 0.2(1, -2)^\top S_t + 2.5R_t\right)$$

where $c_0$ is calibrated to attain target missing rates ranging from 20% to 80%. The target policy we want to evaluate is given by

$$P_\pi(A_t = 1 \mid S_t, O_{t-1}) = \text{expit}\left\{3[(1, 0.3)^\top S_t + 0.5 - 0.8(2O_{t-1} - 1)]\right\}$$

See Figure 4 in Appendix C for visualization of the generated data.

For function classes, we choose Gaussian kernels for $\mathcal{G}^{(t)}$ and $\mathcal{B}^{(t)}$. We report MSE under three settings: varying the sample size, varying the horizon length, and varying the reward generation mechanism. Specifically, we consider

$n \in \{64, 128, 256, 512, 1024, 2048\}$ with $T = 8$, and $T \in \{2, 4, 8, 16, 32\}$ with $n = 512$. More simulation details can be found in Appendix C.1.

Figure 2 shows the MSE of prox (our method), and four baselines naive (naive FQE), ipw (IPW-FQE), impute (Impute-FQE), and scope (SCOPE) under three MNAR missingness levels, approximately 20%, 40%, and 80%. Across all three panels, prox achieves the fastest error decay in $n$ and consistently the smallest MSE, with its advantage growing under heavier missingness. The baselines plateau at non-vanishing bias floors (naive, ipw), amplify selection bias (impute), or degrade under high missingness (scope), consistent with our theoretical analysis. Overall, the results demonstrate that prox is the most accurate and stable estimator in this setting.

Additional simulation results for varying horizon lengths and different reward generation mechanisms are reported in Appendix C in the Supplemental Materials.

## 6.2. Application: MIMIC-III Sepsis data

We evaluate our method on a real-world clinical dataset from the MIMIC-III database (Johnson et al., 2016), using the sepsis cohort and pre-processing pipeline of Raghu et al. (2017). The dataset consists of ICU patients meeting the Sepsis-3 criteria, where physiological measurements, lab values, and treatment records are aggregated into 4-hour windows. Each patient's state at time $t$ is represented by a 48-dimensional feature vector comprising demographics and static indices (e.g., age, SOFA score, SIRS), laboratory values, vital sign, and intake/output variables. The action space consists of 25 discrete treatment options formed by 5 vasopressor dose levels $\times$ 5 IV fluid dose levels, where non-zero dosages are discretized into quartiles.

We retain patients with at least 10 recorded time steps and truncate to the first 10, yielding a horizon of $T = 10$ with 13,943 patients. The reward at each step is defined as $R_t = -(SOFA_{t+1} - SOFA_t)$. Since the MIMIC-III rewards are fully observed, we introduce synthetic MNAR missingness to evaluate our method. The target policy is constructed by training a Double DQN on the fully-observed data and then applying a conservative dose reduction, yielding a missingness-aware policy $\pi(a \mid s_t, o_{t-1})$. Additional setups are provided in Appendix C.4.

We compare prox against oracle (oracle FQE), which uses fully observed rewards as a reference), naive, impute, ipw, and scope.

Figure 3 presents the results (full numerical results are provided in Table 1 in Appendix C.4). prox attains the lowest bias across all missing rates and remains close to oracle even under severe missingness, while all baselines exhibit substantially larger and growing bias as the missing rate

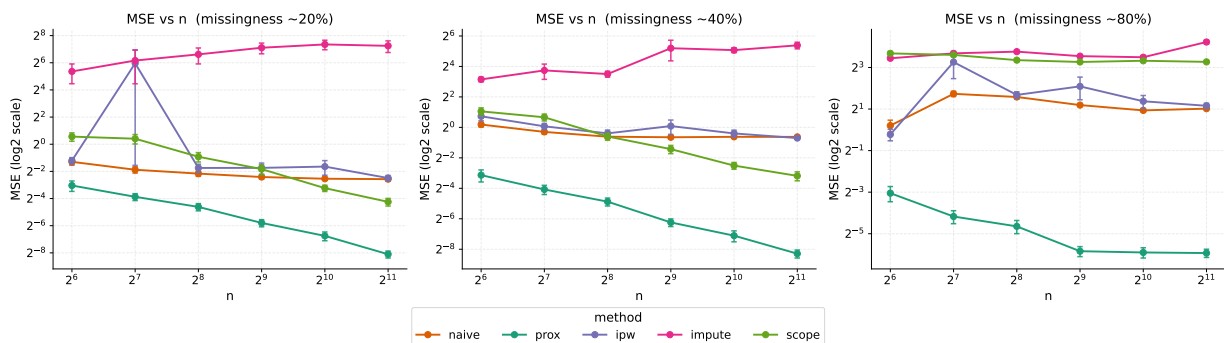

*Figure 2.* MSE vs. sample size ($n$) under three MNAR missingness percentages ($\sim20\%$, $\sim40\%$, $\sim80\%$). `prox` consistently achieves the lowest MSE across all sample sizes and missingness levels, with MSE decreasing steadily as $n$ grows.

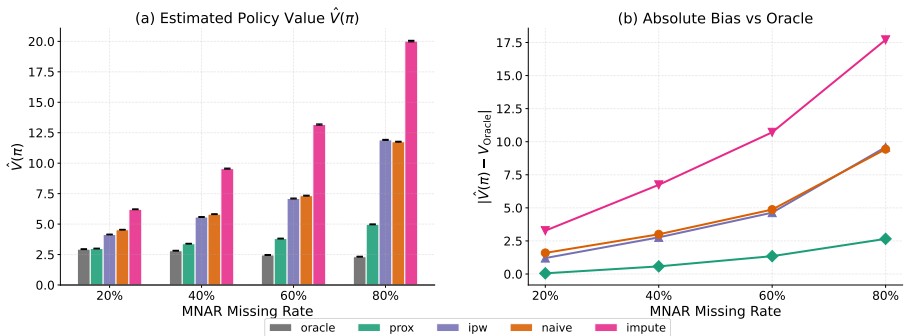

*Figure 3.* OPE results on MIMIC-III sepsis data under MNAR missing rates from 20% to 80%. (a) Estimated policy value $\hat{V}(\pi)$ with standard error bars. oracle FQE (gray) uses fully observed rewards as a reference. (b) Absolute bias relative to oracle FQE. SCOPE is excluded due to degenerate estimates.

increases. `scope` produces degenerate estimates due to near-zero importance weight overlap and is excluded from the figure.

## 7. Conclusion and Discussion

We study OPE in MDPs with MNAR rewards, proposing a bridge function approach that recovers the conditional mean reward without explicitly modeling the missingness mechanism. Unlike IPW-based methods (Wang et al., 2025) that require external auxiliary variables, our method uses the next state as an endogenous shadow variable, avoiding additional data requirements and IPW variance inflation.

A limitation of our framework is that the reward missingness process may still be influenced by unobserved confounding factors that are not fully captured by the observed trajectories. While our identification relies on assumptions through the bridge function, violations of these assumptions may lead to biased policy value estimates in practice.

To address this concern, an important future direction is to incorporate sensitivity analysis into the proposed OPE framework. By parameterizing deviations from the bridge moment conditions or the completeness assumptions, one can assess the robustness of policy value estimates to potential unobserved confounding in the missingness mechanism. Such sensitivity analyses have been studied in proximal causal inference and POMDPs, and adapting their frameworks from proximal causal inference and POMDPs to the reward-missingness mechanism in MDPs is an important direction for future work.

## Acknowledgement

Qu's research is partially supported by NSF Grant DMS 2515275, NCI grant 1R01CA297869, NSF Grant CDS&E-MSS 2401271. Miao's research is partially supported by Texas Artificial Intelligence Institute at University of Texas at Dallas.

## Impact Statement

This paper presents work whose goal is to advance the field of machine learning. There are many potential societal consequences of our work, none of which we feel must be specifically highlighted here.

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

# A. Related Work

**OPE.** Off-policy evaluation has been extensively studied in the RL literature. Classical methods include IS and its variants (Liu et al., 2018), FQE (Le et al., 2019), and doubly robust estimators that combine both (Kallus & Uehara, 2020). Recent advances in offline RL have developed pessimistic approaches (Xie et al., 2021; Rashidinejad et al., 2021; Shi et al., 2023; Zhan et al., 2022) that achieve near-optimal sample complexity. For comprehensive reviews, see Uehara et al. (2022) and Levine et al. (2020).

OPE in POMDPs has received growing attention, with works addressing latent confounding (Bennett et al., 2021; Kallus & Zhou, 2020), partial observability (Tennenholtz et al., 2020; Shi et al., 2022; Miao et al., 2022), and future-dependent estimation (Uehara et al., 2023; Zhang & Jiang, 2024). More recent extensions study OPE of history-dependent policies through model-based methods (Zhang & Jiang, 2025), confounded POMDPs (Kuang et al., 2026), and concept-based representations (Majumdar et al., 2025). These works tackle partial observability in the state process, while our work addresses a complementary challenge: MNAR missingness in the reward process under a fully observed MDP.

**Missing Data.** Missing data problems have been extensively studied in statistics (Little & Rubin, 2019; Enders, 2022). Under missing at random (MAR) assumptions, inverse probability weighting and doubly robust methods are well-established. For MNAR, identification typically requires additional structure such as instrumental variables (Sun & Tchetgen Tchetgen, 2018), shadow variables (Shao & Wang, 2016; Miao & Tchetgen Tchetgen, 2016; Miao et al., 2018), or graphical constraints (Mohan & Pearl, 2021). Proximal causal inference (Tchetgen Tchetgen et al., 2020; Bennett & Kallus, 2021; Cui et al., 2024) has emerged as a powerful framework for handling unmeasured confounding using proxy variables. Our work extends these ideas to the OPE setting with MNAR rewards, leveraging future states as shadow variables for identification.

**Reward Shaping.** Potential-based reward shaping augments rewards with a potential difference to densify sparse feedback while preserving the optimal policy (Ng et al., 1999), with later extensions to dynamic shaping and off-policy settings (Devlin & Kudenko, 2012; Harutyunyan et al., 2015; Parbhoo et al., 2020). These methods are designed to accelerate policy learning or reduce variance under fully observed rewards, whereas our work targets identification under MNAR rewards via a bridge function.

**Semi-supervised RL.** Our setting also relates to semi-supervised RL and partially reward-labeled sequential decision-making, including semi-supervised RL with rewards available only in labeled MDPs (Finn et al., 2016), semi-supervised offline RL with mixed labeled and unlabeled trajectories (Zheng et al., 2023), and offline policy learning with partially reward-labeled trajectories (Li et al., 2023). These works focus on policy learning or reward modeling under label scarcity, whereas we target off-policy *evaluation* under an MNAR missingness mechanism that depends on the latent reward.

# B. Discussion on shadow variables

In the causal inference literature on missing data, identification typically requires additional assumptions, most commonly instantiated through either instrumental variables or shadow variables. Shadow-variable approaches (Miao et al., 2015; Miao & Tchetgen Tchetgen, 2016) leverage a fully observed variable that is informative about the outcome while being conditionally independent of the missingness mechanism given covariates and the (possibly unobserved) outcome. In contrast, instrumental-variable approaches (Sun & Tchetgen Tchetgen, 2018) posit a variable that shifts the missingness mechanism but has no direct effect on the outcome.

In sequential decision making, Wang et al. (2025) develop an approach that requires specifying a stage-wise shadow variable $Z_t$ that satisfies conditional independence with dropout given $(S_t, A_t, R_{t+1}, S_{t+1})$ while remaining informative about $(R_{t+1}, S_{t+1})$ on the observed subset, effectively providing the identifying leverage needed to learn the dropout model.

While such a choice can be plausible, it may rely on additional measurements or domain knowledge to select a valid $Z_t$. In contrast, we adopt an endogenous choice of shadow variable, which is the next state $S_{t+1}$. Under the exclusion restriction and relevance condition in Assumptions 3.1 and 3.2, $S_{t+1}$ provides a readily available proxy for the MNAR reward without introducing an extra auxiliary variable.

Moreover, a natural extension of $S_{t+1}$ is a multi-step future variable, or a low-dimensional summary thereof. This aligns with a broader predictive-state perspective in POMDPs, where future observations are used to encode information about the latent state (Littman & Sutton, 2001; Singh et al., 2003). More recently, Xu et al. (2023); Uehara et al. (2023) use multi-step

futures to stand in for the unobserved state: instead of conditioning on the latent state, they condition on a future window and learn quantities from it, using the future window as a proxy that carries latent-state information.

In missing data problems in MDPs, using longer futures can carry richer information about the missing rewards, and may make relevance and completeness-type conditions more plausible, but it also increases the statistical and computational burden as the future window grows. In practice, these tradeoffs motivate using compact summaries of multi-step futures.

# C. Additional Experiment Details

## C.1. Additional simulation setups

We set the behavior policy that generates the offline trajectories as

$$P_{\pi^b}(A_t = 1 \mid S_t) = \text{expit}\big(0.3 + (0.8, -0.3)^\top S_t\big).$$

The next state $S_{t+1}$ is generated by transition kernel $S_{t+1} = 0.9 S_t + 0.2 A_t \mathbf{1}_2 + \mathcal{N}(0, 0.1^2 I_2)$ where $\mathbf{1}_2 = (1,1)^\top$ and initial state $S_1 \sim \mathcal{N}(0, I_2)$. We consider two reward generation mechanisms. The first is the sigmoid reward model used in other simulations:

$$R_t = \text{expit}([0.9 - 0.6 A_t, -0.7]^\top S_t + [1.3, 2]^\top S_{t+1} - 0.4 A_t) + U_t, \quad U_t \sim \text{Unif}[-0.1, 0.1],$$

and the second is a linear reward model:

$$R_t = \text{clip}([0.5, -0.3]^\top S_t + [0.8, 0.6]^\top S_{t+1} - 0.3 A_t + \epsilon_t, -1, 1), \quad \epsilon_t \sim N(0, 0.01).$$

The true policy value is estimated by Monte Carlo using 5000 independent trajectories generated under the target policy. We visualize the generated data for the setting $n = 1000$, $T = 10$, and random seed 44; see Figure 4 for an overview.

For all the RKHSs, the bandwidths are selected by median heuristic trick (Fukumizu et al., 2009); parameter $\delta$ is set to $\delta_t = 5 n_t^{-0.4}$ according to (Dikkala et al., 2020). The penalty parameter $\lambda_{\text{rkhs}}$ is chosen by cross-validation.

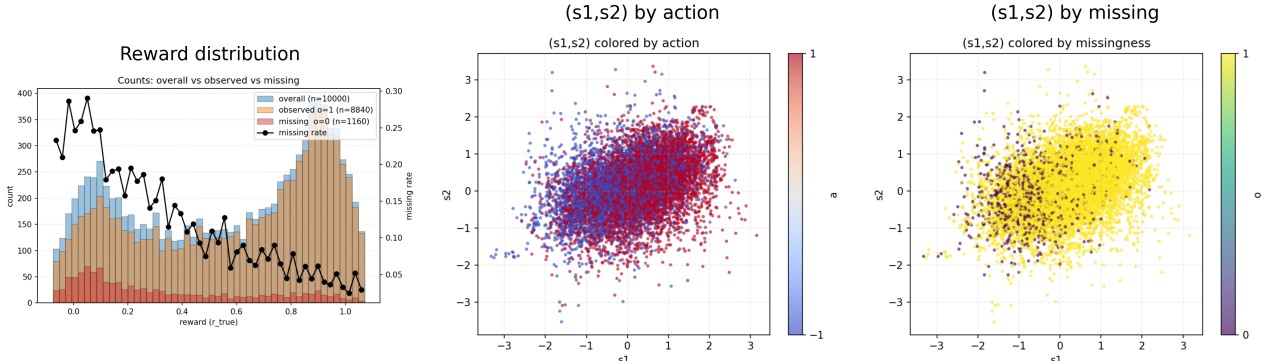

*Figure 4.* **Data overview.** **Left**: histogram of the true reward $r_{\text{true}}$ with three overlays: overall (blue), observed $O{=}1$ (orange), and missing $O{=}0$ (red). The total missing rate is 11.60%. Missing mass is relatively larger in the low–reward region. **Middle**: state value $S_t = (s_1, s_2)$ colored by action (blue: $a{=}-1$, red: $a{=}+1$) according to target policy, showing both actions across the state space without obvious coverage gaps. **Right**: state value $S_t = (s_1, s_2)$ colored by missingness (yellow: $O{=}1$, purple: $O{=}0$); the non-uniform placement of missing points indicates observation probability varies with state.

## C.2. Baseline methods for simulation

### C.2.1. NAIVE FQE

In naive FQE baseline, we ignore the missingness mechanism, and perform FQE only on observed samples:

$$\widehat{Q}_t^{\text{naive}} = \arg \min_{Q \in \mathcal{Q}^{(t)}} \frac{1}{n_t} \sum_{i \in \mathcal{I}_t^{\text{obs}}} \Big( Q(S_{t,i}, A_{t,i}) - y_{t,i}^{\text{naive}} \Big)^2 + \lambda_{\text{rkhs}} \|Q\|_{\mathcal{Q}^{(t)}}^2,$$

where

$$
y_{t,i}^{\text{naive}} = \begin{cases} R_{t,i}, & t = T, \\ R_{t,i} + \sum_{a \in \mathcal{A}} \pi(a \mid S_{t+1,i}, O_{t,i}) \, \widehat{Q}_{t+1}^{\text{naive}}(S_{t+1,i}, a), & t < T. \end{cases}
$$

As discussed in Section 3, under reward MNAR we generally have

$$
\mathbb{E}[R_t \mid S_t = s, A_t = a, O_t = 1] \neq \mathbb{E}[R_t \mid S_t = s, A_t = a] \quad \text{for some } (s, a),
$$

so naive FQE, which regresses on observed rewards, can be biased.

### C.2.2. IPW-FQE

We adapt the IPW-based FQE method of Wang et al. (2025) to our reward MNAR setting. Since the true reward $R_t$ is unobserved when $O_t = 0$, we cannot directly condition the propensity score on $R_t$. Instead, we first fit an RKHS bridge function $\hat{b}_t$ on observed samples ($O_t = 1$) to obtain $\widetilde{R}_t$, and then estimate the extended propensity score via logistic regression on features $(S_t, A_t, \hat{b}_t)$:

$$
\hat{e}_t(S_t, A_t, \hat{b}_t) = \hat{P}(O_t = 1 \mid S_t, A_t, \hat{b}_t).
$$

At each stage $t$, IPW-FQE solves a weighted kernel ridge regression on the observed subset $\mathcal{I}_t^{\text{obs}}$:

$$
\widehat{Q}_t^{\text{ipw}} = \arg \min_{Q \in \mathcal{Q}^{(t)}} \frac{1}{n_t} \sum_{i \in \mathcal{I}_t^{\text{obs}}} w_{t,i} \Big( Q(S_{t,i}, A_{t,i}) - y_{t,i}^{\text{ipw}} \Big)^2 + \lambda_{\text{rkhs}} \|Q\|_{\mathcal{Q}^{(t)}}^2,
$$

where $w_{t,i} = \frac{1}{\hat{e}_{t,i}}$ and

$$
y_{t,i}^{\text{ipw}} = \begin{cases} R_{t,i}, & t = T, \\ R_{t,i} + \sum_{a \in \mathcal{A}} \pi(a \mid S_{t+1,i}, O_{t,i}) \, \widehat{Q}_{t+1}^{\text{ipw}}(S_{t+1,i}, a), & t < T. \end{cases}
$$

### C.2.3. IMPUTE-FQE

At each step $t$, we fit a kernel ridge regression on the observed subset to learn the conditional mean of the reward function $m_t(S_t, A_t)$. The imputed rewards are then constructed as

$$
\widetilde{R}_{t,i}^{\text{imp}} = O_{t,i} \cdot R_{t,i} + (1 - O_{t,i}) \cdot \hat{m}_t(S_{t,i}, A_{t,i}),
$$

and FQE proceeds on all $n_t$ samples using $\widetilde{R}_{t,i}^{\text{imp}}$ in place of $R_{t,i}$:

$$
\widehat{Q}_t^{\text{imp}} = \arg \min_{Q \in \mathcal{Q}^{(t)}} \frac{1}{n} \sum_{i=1}^{n} \Big( Q(S_{t,i}, A_{t,i}) - y_{t,i}^{\text{imp}} \Big)^2 + \lambda_{\text{rkhs}} \|Q\|_{\mathcal{Q}^{(t)}}^2,
$$

where

$$
y_{t,i}^{\text{imp}} = \begin{cases} \widetilde{R}_{t,i}^{\text{imp}}, & t = T, \\ \widetilde{R}_{t,i}^{\text{imp}} + \sum_{a \in \mathcal{A}} \pi(a \mid S_{t+1,i}, O_{t,i}) \, \widehat{Q}_{t+1}^{\text{imp}}(S_{t+1,i}, a), & t < T. \end{cases}
$$

Since $\hat{m}_t$ is trained only on the observed subset $\{i : O_{t,i} = 1\}$, it estimates $\mathbb{E}[R_t \mid S_t, A_t, O_t = 1]$ rather than $\mathbb{E}[R_t \mid S_t, A_t]$. Under MNAR, the imputed values inherit the selection bias from the observed subsample, leading to biased Q-function estimates.

### C.2.4. SCOPE

SCOPE (Parbhoo et al., 2020) is a per-step importance sampling estimator designed for sparse reward settings, which incorporates potential-based reward shaping (Ng et al., 1999) as a control variate to reduce variance. To apply SCOPE in our MNAR setting, we replace unobserved rewards with zero. This introduces bias because under MNAR, missingness is informative and zero-imputation does not recover the true expected reward.

## C.3. Other simulation results

Figure 5 reports MSE as the horizon $T$ varies from 2 to 32. Error compounding in backward induction affects all methods, but the growth rate differs markedly. `prox` remains below MSE $\approx 1$ for $T \leq 16$ across all missingness levels, whereas `ipw` already exceeds $10^2$ at $T = 16$ under $\sim 40\%$ missingness. `impute` is the worst-performing method at longer horizons, with MSE exceeding $10^8$ at $T = 32$ under $\sim 20\%$ missingness, because imputation errors at each step feed into subsequent Q-function fits and amplify exponentially. `scope` is the most competitive baseline for short horizons ($T \leq 4$) but its per-step IS weights accumulate variance as $T$ grows. These results highlight that the bridge-function approach in `prox` is particularly advantageous in long-horizon problems.

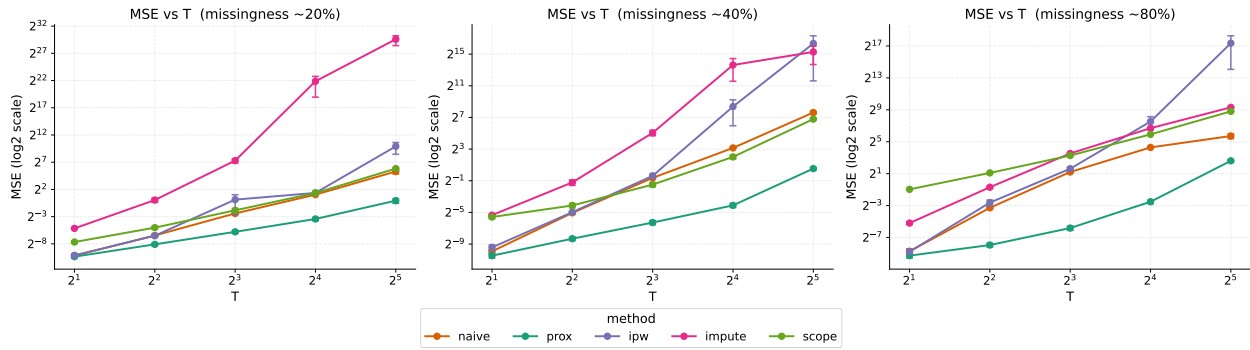

*Figure 5.* MSE vs. horizon length ($T$) under three MNAR missingness percentages ($\sim 20\%, \sim 40\%, \sim 80\%$). `prox` scales most gracefully as $T$ grows, maintaining orders-of-magnitude lower MSE than all baselines. `ipw` and `impute` explode at large $T$ due to variance blow-up and compounding imputation errors, respectively. `naive` grows steadily as MNAR bias compounds at each backward step. `scope` remains more stable than `ipw` but degrades sharply under high missingness.

For the reward type comparison, we fix $n = 512$ and $T = 8$ and evaluate under the two reward generation mechanisms: a bounded sigmoid reward and a clipped linear reward. Figure 6 compares MSE across the two reward generation mechanisms. Under the sigmoid reward, `prox` achieves MSE on the order of $10^{-2}$, while the best baseline (`naive`) is above $10^{-1}$ even at $\sim 20\%$ missingness. The nonlinearity of the sigmoid function makes regression-based imputation particularly difficult, explaining the poor performance of `impute` on this reward type. Under the linear reward, all methods improve and the gap between `prox` and baselines narrows. Notably, `impute` and `scope` become competitive with `prox` at $\sim 80\%$ missingness for the linear reward, suggesting that simpler reward structures are more amenable to naive correction strategies.

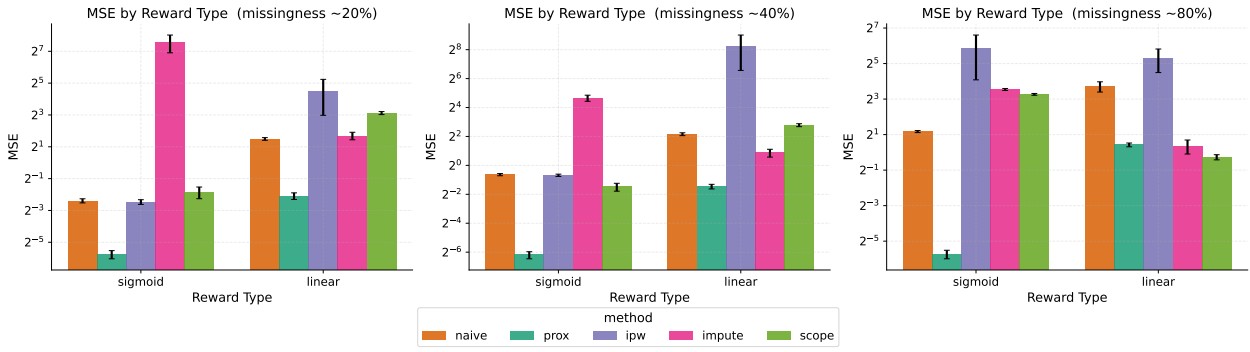

*Figure 6.* MSE by reward type (sigmoid vs. linear) under three MNAR missingness percentages ($\sim 20\%, \sim 40\%, \sim 80\%$). For the sigmoid reward, `prox` achieves MSE orders of magnitude lower than all baselines. For the linear reward, the gap narrows but `prox` still leads. `impute` performs comparably on the linear reward but poorly on sigmoid. `scope` degrades sharply under high missingness for sigmoid.

## C.4. Additional real data experiment setups

The propensity score is set as

$$e_t(S_t, A_t, R_t) = \text{expit}(c_0 + 2.0 \cdot R_t^{\text{std}} + 0.3 \cdot S_t^{\text{signal}} - 0.1 \cdot A_t^{\text{norm}}),$$

where $R_t^{\text{std}} = (R_t - \bar{R})/\text{sd}(R)$ is the standardized reward, $A_t^{\text{norm}}$ is the standardized total treatment intensity (sum of vasopressor and IV fluid dose levels, then z-scored), and $S_t^{\text{signal}}$ is a clinical severity signal defined as the normalized arterial lactate minus half the normalized mean blood pressure. The positive coefficient on the reward ($c_r = 2.0$) ensures that higher rewards are more likely to be observed while lower rewards tend to be missing, reflecting a clinically plausible scenario where SOFA-based outcomes depend on lab measurements that may be ordered less frequently for deteriorating patients. The intercept $c_0$ is calibrated via bisection to achieve target missing rates of 20%, 40%, 60%, and 80%.

The target policy is constructed in two stages. First, we train a Double DQN with two hidden layers of size 128 on the original fully-observed data using the 48-dimensional state and $\gamma = 1.0$. This produces a greedy base policy $\pi_{\text{DQN}}(s_t)$ over the 25 actions. Second, we construct a missingness-aware policy $\pi(a \mid s_t, o_{t-1})$ by applying a conservative dose reduction when the previous reward is unobserved: if $O_{t-1} = 0$, both the vasopressor and IV fluid dose levels recommended by the DQN are reduced by one (clipped at zero). This adjustment reflects a clinically natural response—reducing treatment intensity when the previous outcome is unknown—and produces a target policy whose action distribution depends on $O_{t-1}$, consistent with the policy class in our framework.

We split patients 60/40 by ID for model fitting and evaluation, respectively. On the fitting set, we train all OPE models including bridge functions, Q-networks, and auxiliary models for the baselines. Due to the high-dimensional state space and large action space, all methods use neural network function approximation, with architectures of $(512, 512, 256)$ for Q-networks and bridge networks and $(256, 256)$ for auxiliary networks (imputation regressor, propensity classifier). On the held-out 40% test set, we compute the per-patient estimated value $\hat{V}_i(\pi)$ and report the mean and standard error. See detailed experiment results in Table 1.

*Table 1.* OPE results on MIMIC-III sepsis data. We report the estimated policy value $\hat{V}(\pi)$ with standard errors and absolute bias relative to oracle FQE.

| Method | 20% MNAR | | 40% MNAR | | 60% MNAR | | 80% MNAR | |
|---|---|---|---|---|---|---|---|---|
| | $\hat{V}(\pi)$ | Bias | $\hat{V}(\pi)$ | Bias | $\hat{V}(\pi)$ | Bias | $\hat{V}(\pi)$ | Bias |
| oracle | $2.94 \pm 0.03$ | — | $2.81 \pm 0.03$ | — | $2.46 \pm 0.03$ | — | $2.32 \pm 0.03$ | — |
| Prox | $2.99 \pm 0.03$ | 0.05 | $3.39 \pm 0.03$ | 0.58 | $3.81 \pm 0.03$ | 1.35 | $4.98 \pm 0.03$ | 2.66 |
| IPW | $4.15 \pm 0.03$ | 1.20 | $5.58 \pm 0.03$ | 2.77 | $7.10 \pm 0.03$ | 4.64 | $11.91 \pm 0.04$ | 9.59 |
| naive | $4.54 \pm 0.03$ | 1.59 | $5.81 \pm 0.03$ | 3.00 | $7.33 \pm 0.03$ | 4.87 | $11.76 \pm 0.03$ | 9.44 |
| impute | $6.22 \pm 0.03$ | 3.27 | $9.55 \pm 0.04$ | 6.74 | $13.17 \pm 0.04$ | 10.71 | $20.02 \pm 0.06$ | 17.70 |
| scope | $10408.2 \pm 1991.0$ | 10405.3 | $-4089.2 \pm 786.8$ | 4092.0 | $10667.9 \pm 2037.3$ | 10665.4 | $-7096.6 \pm 1373.6$ | 7099.0 |

# D. Proof of Theorem 3.6

In this section we provide the proof of identification result in Theorem 3.6.

**Part A: Identification by bridge functions**

We first show that the policy value can be identified by the bridge functions $\{b_t\}_{t=1}^T$ if the bridges exist.

Fix $t \in \{1, \ldots, T\}$ and suppose there exists a measurable function $b_t : \mathcal{S} \times \mathcal{A} \times \mathcal{S} \to \mathbb{R}$ satisfying the Equation (2). By Assumptions 2.1 and 3.3, conditioning on the event $O_t = 1$ is well-defined and

$$\mathbb{E}[b_t(S_t, A_t, S_{t+1}) \mid R_t, S_t, A_t, O_t = 1] = \mathbb{E}[b_t(S_t, A_t, S_{t+1}) \mid R_t, S_t, A_t].$$

So Equation (2) holds if and only if Equation (6) holds.

For the imputed reward $\widetilde{R}_t$ defined by Equation (4), Equation (5) implies that it has the same conditional mean reward as $R_t$ given $(S_t, A_t)$:

$$\mathbb{E}[\widetilde{R}_t \mid S_t = s, A_t = a] = \mathbb{E}[b_t(s, a, S_{t+1}) \mid S_t = s, A_t = a] = \mathbb{E}[R_t \mid S_t = s, A_t = a] := \bar{r}_t(s, a).$$

Now consider the augmented process. By Assumption 2.2, the augmented process is Markov, and the Bellman recursion holds with one-step reward $\bar{r}_t(S_t, A_t)$:

$$Q_t^\pi(s,a) = \mathbb{E}\big[\bar{r}_t(s,a) + V_{t+1}^\pi(S_{t+1}, O_t) \mid S_t = s, A_t = a\big],$$

$$V_t^\pi(s, o_-) = \sum_a \pi_t(a \mid s, o_-)Q_t^\pi(s,a), \qquad V_{T+1}^\pi \equiv 0,$$

which is equivalent to Equation (1). Therefore, the policy value $V(\pi) = \mathbb{E}[V_1^\pi(\widetilde{S}_1)]$ is identified.

**Part B: Existence of bridge functions**

We now establish existence of the bridges. For a probability measure $\mu$, let $\mathcal{L}^2(\mu)$ denote the space of all squared integrable functions of $x$ with respect to measure $\mu(x)$, which is a Hilbert space endowed with the inner product $\langle g_1, g_2 \rangle = \int g_1(x)g_2(x)d\mu(x)$. For any $s, a, t$, we define operator

$$\mathcal{T}_{t|(s,a)} : \mathcal{L}^2\big(P_{S_{t+1}|s,a}\big) \to \mathcal{L}^2\big(P_{R_t|s,a}\big),$$

where $(\mathcal{T}_{t|(s,a)}h)(r) := \mathbb{E}\big[h(S_{t+1}) \mid R_t = r, S_t = s, A_t = a\big]$. Its adjoint operator is defined by

$$\mathcal{T}_{t|(s,a)}^* : \mathcal{L}^2\big(P_{R_t|s,a}\big) \to \mathcal{L}^2\big(P_{S_{t+1}|s,a}\big),$$

where $(\mathcal{T}_{t|(s,a)}^* g)(s') = \mathbb{E}\big[g(R_t) \mid S_{t+1} = s', S_t = s, A_t = a\big]$. Then, the bridge equation (2) can be written as a first-kind Fredholm integral equation

$$(\mathcal{T}_{t|(s,a)}h)(r) = g(r),$$

where the unknown functions are $h(\cdot) = b_t(s,a,\cdot) \in \mathcal{L}^2\big(P_{S_{t+1}|s,a}\big)$ and the right hand side is $g(r) = r \in \mathcal{L}^2(P_{R_t|s,a})$.

**Assumption D.1** (Hilbert-Schmidt property). For any $(s,a) \in \mathcal{S} \times \mathcal{A}$, for all $t = 1, \ldots, T$, denote conditional densities $p_{S_{t+1}|R_t}(s' \mid r, s, a)$, $p_{R_t|S_{t+1}}(r \mid s', s, a)$. We have

$$\int_{\mathcal{R}} \int_{\mathcal{S}} p_{S_{t+1}|R_t}(s' \mid r, s, a) p_{R_t|S_{t+1}}(r \mid s', s, a)ds'dr < \infty.$$

This ensures that the operator $\mathcal{T}_{t|(s,a)}$ is Hilbert–Schmidt and thus admits a singular system $\{(\sigma_{s,a,t,\nu}, \varphi_{s,a,t,\nu}, \psi_{s,a,t,\nu})\}_{\nu \geq 1}$ satisfying $\mathcal{T}_{t|(s,a)}\varphi_{s,a,t,\nu} = \sigma_{s,a,t,\nu}\psi_{s,a,t,\nu}$ and $\mathcal{T}_{t|(s,a)}^*\psi_{s,a,t,\nu} = \sigma_{s,a,t,\nu}\varphi_{s,a,t,\nu}$.

**Assumption D.2.** Suppose $\{(\sigma_{s,a,t,\nu}, \varphi_{s,a,t,\nu}, \psi_{s,a,t,\nu})\}_{\nu \geq 1}$ is a singular system of $\mathcal{T}_{t|(s,a)}$. Then for all $(s,a) \in \mathcal{S} \times \mathcal{A}$ and $t = 1, \ldots, T$,

$$\sum_{\nu \geq 1} \frac{\langle g, \psi_{s,a,t,\nu} \rangle_{\mathcal{L}^2(P_{R_t|s,a})}^2}{\sigma_{s,a,t,\nu}^2} < \infty,$$

where $\langle g, \psi_{s,a,t,\nu} \rangle_{\mathcal{L}^2(P_{R_t|s,a})} = \int_{\mathcal{R}} g(r) \cdot \psi_{s,a,t,\nu}(r)p_{R_t|s,a}(r)dr$.

**Lemma D.3** (Picard's Theorem (Kress, 1989)). *Let $\mathcal{H}_1, \mathcal{H}_2$ be real Hilbert spaces and $K : \mathcal{H}_1 \to \mathcal{H}_2$ a compact linear operator with adjoint $K^* : \mathcal{H}_2 \to \mathcal{H}_1$. Then, there exists a singular system $\{(\lambda_\nu, \phi_\nu, \psi_\nu)\}_{\nu=1}^\infty$ of $K$, with singular values $\lambda_\nu > 0$ and orthonormal sequences $\{\phi_\nu\} \subset \mathcal{H}_1$, $\{\psi_\nu\} \subset \mathcal{H}_2$ satisfying*

$$K\phi_\nu = \lambda_\nu \psi_\nu, \quad K^*\psi_\nu = \lambda_\nu \phi_\nu.$$

*Given $g \in \mathcal{H}_2$, the first–kind Fredholm equation $Kh = g$ has a solution $h \in \mathcal{H}_1$ if and only if*

*(a) $g \in \ker(K^*)^\perp$;*

*(b) $\sum_{\nu=1}^\infty \lambda_\nu^{-2}|\langle g, \psi_\nu \rangle_{\mathcal{H}_2}|^2 < \infty$,*

*where $\ker(K^*) = \{h : K^*h = 0\}$ is the null space of $K^*$, and $\perp$ denotes the orthogonal complement to a set.*

**Proposition D.4** (Existence of bridges). *Under Assumption 3.4 (1), Assumptions D.1 and D.2, for all $(s,a) \in \mathcal{S} \times \mathcal{A}$ and $t = 1, \ldots, T$, there exists a solution $h$ to equation*

$$\mathcal{T}_{t|(s,a)} h = g,$$

*where $h := b_t(s,a; \cdot) \in \mathcal{L}^2\big(P_{S_{t+1}|s,a}\big)$ and $g(r) = r \in \mathcal{L}^2(P_{R_t|s,a})$. Equivalently, for any fixed $(s,a)$ and $t$, there exists a function $b_t(s, a, \cdot)$ satisfying Equation (2).*

*Proof.* By Assumption D.1, for all $(s,a) \in \mathcal{S} \times \mathcal{A}$ and $t = 1, \ldots, T$, the operator $\mathcal{T}_{t|(s,a)}$ is Hilbert–Schmidt and thus compact. Suppose there exists function $f \in \ker(\mathcal{T}^*_{t|(s,a)})$, then by definition, $\mathcal{T}^*_{t|(s,a)} f = 0$. By Assumption 3.4 (1), we have $f(R_t) = 0$, *a.s.* Therefore, $\ker(\mathcal{T}^*_{t|(s,a)}) = \{0\}$, and hence $\ker(\mathcal{T}^*_{t|(s,a)})^\perp = \mathcal{L}^2\big(P_{R_t|s,a}\big)$. Because reward $R_t$ is bounded, then $g \in \mathcal{L}^2\big(P_{R_t|s,a}\big)$. So the condition (a) in Lemma D.3 is satisfied.

Additionally, condition (b) is also satisfied by Assumption D.2. By Lemma D.3, there exists a solution $h \in \mathcal{L}^2(P_{S_{t+1}|s,a})$ to $\mathcal{T}_{t|(s,a)} h = g$ where $g(r) = r$, i.e., there exists function $b_t(s, a, \cdot)$ such that:

$$\mathbb{E}\big[b_t(s, a, S_{t+1}) \mid R_t = r, S_t = s, A_t = a\big] = r.$$

$\square$

**Part C: Uniqueness of bridge functions**

Next we study the uniqueness of the bridge functions.

**Proposition D.5** (Uniqueness of bridges). *Under Assumption 3.4 (2), any bridge function $b_t$ satisfying*

$$\mathbb{E}[b_t(S_t, A_t, S_{t+1}) \mid R_t, S_t, A_t] = R_t, \quad a.s.$$

*is unique.*

*Proof.* Suppose there exist different bridge functions $b_{1,t}$ and $b_{2,t}$ satisfying the equation above for any $t$. Then,

$$\mathbb{E}[b_{1,t}(S_t, A_t, S_{t+1}) - b_{2,t}(S_t, A_t, S_{t+1}) \mid R_t, S_t, A_t] = 0, \quad a.s.$$

By Assumption 3.4 (2), we have

$$b_{1,t} - b_{2,t} = 0, \quad a.s., \quad \forall t = 1, \ldots, T,$$

which contradicts $b_{1,t} \neq b_{2,t}$. Thus uniqueness holds. $\square$

# E. Proof of Theorem 5.5

For a function class $\mathcal{G}$ and radius $\delta > 0$, given sample $\{X_i\}$, the *local empirical Rademacher complexity* is defined by

$$\widehat{\mathcal{R}}_n(\mathcal{G}, \delta) = \mathbb{E}_\varepsilon\Big[ \sup_{g \in \mathcal{G}: \|g\|_{2,n} \leq \delta} \big|\frac{1}{n} \sum_{i=1}^n \varepsilon_i g(X_i)\big| \,\Big|\, \{X_i\}\Big]$$

where $\|g\|_{2,n}^2 = \frac{1}{n} \sum_{i=1}^n g(X_i)^2$. The empirical critical radius $\hat{\delta}_n$ is the smallest solution to the inequality $\widehat{\mathcal{R}}_n(\mathcal{G}, \delta) \leq \frac{\delta^2}{b}$. Wainwright (2019) gives the relationship of critical radius and empirical critical radius: with probability at least $1 - \zeta$,

$$\delta_n \leq \mathcal{O}(\hat{\delta}_n + \sqrt{\frac{\log(1/\zeta)}{n}}),$$

which enables us to study on empirical critical radius $\hat{\delta}_n$.

Define

$$\Psi_n^t(b, g) = \frac{1}{n_t} \sum_{i \in \mathcal{I}_t^{\mathrm{obs}}} (b_t(S_{t,i}, A_{t,i}, S_{t+1,i}) - R_{t,i}^{\mathrm{obs}}) g_t(R_{t,i}^{\mathrm{obs}}, S_{t,i}, A_{t,i}),$$

and population level version

$$\Psi^t(b, g) = \mathbb{E}\Big[(b_t(S_t, A_t, S_{t+1}) - R_t)g_t(R_t, S_t, A_t) \,|\, O_t = 1\Big].$$

Moreover, Let

$$\Psi_n^{t,\lambda}(b, g) = \Psi_n^t(b, g) - \lambda\left(\|g_t\|_{\mathcal{G}^{(t)}}^2 + \frac{U}{\delta^2}\|g_t\|_{2,n_t}^2\right),$$

$$\Psi^{t,\lambda}(b, g) = \Psi^t(b, g) - \lambda\left(\frac{2}{3}\|g_t\|_{\mathcal{G}^{(t)}}^2 + \frac{U}{2\delta^2}\|g_t\|_2^2\right).$$

So the minimizer $\hat{b}_t$ can be written as

$$\hat{b}_t = \arg\min_{b_t \in \mathcal{B}^{(t)}} \sup_{g_t \in \mathcal{G}^{(t)}} \Psi_n^{t,\lambda}(b, g) + \lambda\mu\|b_t\|_{\mathcal{B}^{(t)}}^2.$$

By Lemma H.2, with probability at least $1 - \zeta$, for any function $g_t \in \mathcal{G}_{3U}^{(t)}$,

$$\left|\|g_t\|_{2,n_t}^2 - \|g_t\|_2^2\right| \le \frac{1}{2}\|g_t\|_2^2 + (\delta_t^{\mathcal{G}})^2,$$

where $\delta_t^{\mathcal{G}} = \delta_{n_t}^{\mathcal{G}} + c_0\sqrt{\frac{\log(c_1/\zeta)}{n_t}}$, and $\delta_{n_t}^{\mathcal{G}}$ is the upper bound of the empirical critical radii of function class $\mathcal{G}_{3U}^{(t)}$. For any $\|g_t\|_{\mathcal{G}^{(t)}}^2 \ge 3U$, consider rescaling $g_t$ by $\frac{\sqrt{3U}}{\|g_t\|_{\mathcal{G}^{(t)}}}g_t \in \mathcal{G}_{3U}^{(t)}$, and we have

$$\left|\|g_t\|_{2,n_t}^2 - \|g_t\|_2^2\right| \le \frac{1}{2}\|g_t\|_2^2 + (\delta_t^{\mathcal{G}})^2\frac{\|g_t\|_{\mathcal{G}^{(t)}}^2}{3U}.$$

Combine the above inequalities,

$$\left|\|g_t\|_{2,n_t}^2 - \|g_t\|_2^2\right| \le \frac{1}{2}\|g_t\|_2^2 + (\delta_t^{\mathcal{G}})^2 \max\left\{1, \frac{\|g_t\|_{\mathcal{G}^{(t)}}^2}{3U}\right\}. \tag{9}$$

Thus,

$$\|g_t\|_{\mathcal{G}^{(t)}}^2 + \frac{U}{\delta^2}\|g_t\|_{2,n_t}^2 \ge \|g_t\|_{\mathcal{G}^{(t)}}^2 + \frac{U}{(\delta_t^{\mathcal{G}})^2}\left[\frac{1}{2}\|g_t\|_2^2 - (\delta_t^{\mathcal{G}})^2 \max\left\{1, \frac{\|g_t\|_{\mathcal{G}^{(t)}}^2}{3U}\right\}\right]$$
$$\ge \frac{2}{3}\|g_t\|_{\mathcal{G}^{(t)}}^2 + \frac{U}{2(\delta_t^{\mathcal{G}})^2}\|g_t\|_2^2 - U. \tag{10}$$

Next, we study the upper and lower bounds of the centered empirical sup-loss

$$\sup_{g_t \in \mathcal{G}^{(t)}} \Psi_n^t(\hat{b}_t, g) - \Psi_n^t(b_t^*, g) - 2\lambda\left(\|g_t\|_{\mathcal{G}^{(t)}}^2 + \frac{U}{(\delta_t^{\mathcal{G}})^2}\|g_t\|_{2,n_t}^2\right).$$

For simplicity we omit $t$ and write it as

$$\sup_{g \in \mathcal{G}} \Psi_n(\hat{b}, g) - \Psi_n(b^*, g) - 2\lambda\left(\|g\|_{\mathcal{G}}^2 + \frac{U}{\delta^2}\|g\|_{2,n}^2\right). \tag{11}$$

### E.1. Upper bounding the centered empirical sup-loss

We first decompose $\Psi_n^\lambda(b, g)$ by

$$\Psi_n^\lambda(b, g) = \Psi_n(b, g) - \Psi_n(b^*, g) + \Psi_n(b^*, g) - \lambda\left(\|g\|_{\mathcal{G}}^2 + \frac{U}{\delta^2}\|g\|_{2,n}^2\right)$$
$$\ge \Psi_n(b, g) - \Psi_n(b^*, g) - 2\lambda\left(\|g\|_{\mathcal{G}}^2 + \frac{U}{\delta^2}\|g\|_{2,n}^2\right) - \sup_{g \in \mathcal{G}} \Psi_n^\lambda(b^*, g),$$

where the last inequality holds by symmetry of $\mathcal{G}$. Then we have

$$
\begin{aligned}
\sup_{g \in \mathcal{G}} \Psi_n(\hat{b}, g) - \Psi_n(b^*, g) - 2\lambda \left( \|g\|_{\mathcal{G}}^2 + \frac{U}{\delta^2} \|g\|_{2,n}^2 \right) &\leq \sup_{g \in \mathcal{G}} \Psi_n^\lambda(b^*, g) + \Psi_n^\lambda(\hat{b}, g) \\
&\leq \sup_{g \in \mathcal{G}} \Psi_n^\lambda(b^*, g) + \left[ \sup_{g \in \mathcal{G}} \Psi_n^\lambda(b^*, g) + \lambda\mu\left( \|b^*\|_{\mathcal{B}}^2 - \|\hat{b}\|_{\mathcal{B}}^2 \right) \right] \quad (12) \\
&\leq 2 \sup_{g \in \mathcal{G}} \Psi_n^\lambda(b^*, g) + \lambda\mu\left( \|b^*\|_{\mathcal{B}}^2 - \|\hat{b}\|_{\mathcal{B}}^2 \right).
\end{aligned}
$$

By Lemma H.3, for all $g \in \mathcal{G}$, similarly, with probability at least $1 - \zeta$ we have

$$
\begin{aligned}
\left| \Psi_n(b^*, g) - \Psi(b^*, g) \right| &\leq 36\delta \left[ \|g\|_2 + \delta \max \left\{ 1, \frac{\|g\|_{\mathcal{G}}}{\sqrt{3U}} \right\} \right] \\
&\leq 36\delta \left[ \|g\|_2 + \delta(1 + \frac{\|g\|_{\mathcal{G}}}{\sqrt{3U}}) \right].
\end{aligned}
\tag{13}
$$

Combine Equation (10) and Equation (13), we have with probability at least $1 - 2\zeta$, for all $b \in \mathcal{B}$ and $g \in \mathcal{G}$,

$$
\begin{aligned}
\Psi_n^\lambda(b^*, g) &= \Psi_n(b^*, g) - \lambda \left( \|g\|_{\mathcal{G}}^2 + \frac{U}{\delta^2} \|g\|_{2,n}^2 \right) \\
&\leq \Psi(b^*, g) + |\Psi_n(b^*, g) - \Psi(b^*, g)| - \lambda \left( \|g\|_{\mathcal{G}}^2 + \frac{U}{\delta^2} \|g\|_{2,n}^2 \right) \\
&\leq \Psi(b^*, g) + 36\delta \left[ \|g\|_2 + \delta(1 + \frac{\|g\|_{\mathcal{G}}}{\sqrt{3U}}) \right] - \lambda \left( \|g\|_{\mathcal{G}}^2 + \frac{U}{\delta^2} \|g\|_{2,n}^2 \right) \\
&\leq \Psi(b^*, g) + 36\delta \left[ \|g\|_2 + \delta(1 + \frac{\|g\|_{\mathcal{G}}}{\sqrt{3U}}) \right] - \lambda \left( \frac{2}{3} \|g\|_{\mathcal{G}}^2 + \frac{U}{2\delta^2} \|g\|_{2,n}^2 \right) + \lambda U \\
&\leq \Psi(b^*, g) - \lambda \left( \frac{1}{3} \|g\|_{\mathcal{G}}^2 + \frac{U}{4\delta^2} \|g\|_{2,n}^2 \right) + 36\delta^2 + \lambda U + 36\delta \|g\|_2 + 36\delta^2 \frac{\|g\|_{\mathcal{G}}}{\sqrt{3U}} - \lambda \frac{U}{4\delta^2} \|g\|_{2,n}^2 - \frac{\lambda}{3} \|g\|_{\mathcal{G}}^2 \\
&= \Psi^{\lambda/2}(b^*, g) + 36\delta^2 + \lambda U + \left( 36\delta \|g\|_2 - \lambda \frac{U}{4\delta^2} \|g\|_{2,n}^2 \right) + \left( 36\delta^2 \frac{\|g\|_{\mathcal{G}}}{\sqrt{3U}} - \frac{\lambda}{3} \|g\|_{\mathcal{G}}^2 \right).
\end{aligned}
$$

Using the fact that for any $a, b > 0$ and any norm, $\sup_{g \in \mathcal{G}} (a\|g\| - b\|g\|^2) \leq \frac{a^2}{4b}$, suppose $\lambda \geq \frac{C_1 \delta^2}{U}$, and then

$$
\begin{aligned}
\sup_{g \in \mathcal{G}} \left( 36\delta \|g\|_2 - \lambda \frac{U}{4\delta^2} \|g\|_{2,n}^2 \right) &\leq \frac{36^2 \delta^4}{\lambda U} \leq \frac{36^2 \delta^2}{C_1}; \\
\sup_{g \in \mathcal{G}} \left( 36\delta^2 \frac{\|g\|_{\mathcal{G}}}{\sqrt{3U}} - \frac{\lambda}{3} \|g\|_{\mathcal{G}}^2 \right) &\leq \frac{18^2 \delta^4}{\lambda U} \leq \frac{18^2 \delta^2}{C_1}.
\end{aligned}
$$

Therefore,

$$
\Psi_n^\lambda(b^*, g) \leq \Psi^{\lambda/2}(b^*, g) + 36\delta^2 + \lambda U + \frac{5 \times 18^2 \delta^2}{C_1}
$$

Combine this with Equation (12) we get

$$
\begin{aligned}
\sup_{g \in \mathcal{G}} \Psi_n(\hat{b}, g) - \Psi_n(b^*, g) - 2\lambda \left( \|g\|_{\mathcal{G}}^2 + \frac{U}{\delta^2} \|g\|_{2,n}^2 \right) &\leq 2 \sup_{g \in \mathcal{G}} \Psi_n^\lambda(b^*, g) + \lambda\mu\left( \|b^*\|_{\mathcal{B}}^2 - \|\hat{b}\|_{\mathcal{B}}^2 \right) \\
&\leq 2 \sup_{g \in \mathcal{G}} \Psi^{\lambda/2}(b^*, g) + 2\lambda U + (72 + \frac{10 \times 18^2}{C_1})\delta^2 + \lambda\mu\left( \|b^*\|_{\mathcal{B}}^2 - \|\hat{b}\|_{\mathcal{B}}^2 \right) \\
&= 2\lambda U + (72 + \frac{10 \times 18^2}{C_1})\delta^2 + \lambda\mu\left( \|b^*\|_{\mathcal{B}}^2 - \|\hat{b}\|_{\mathcal{B}}^2 \right).
\end{aligned}
\tag{14}
$$

## E.2. Lower bounding the centered empirical sup-loss

By Assumption 5.4, we write $g_b = \arg\inf_{g \in \mathcal{G}_{L^2 \|b - b^*\|_{\mathcal{B}}^2}} \|g - \mathcal{T}(b - b^*)\|_2$ where

$$\sup_{b \in \mathcal{B}} \inf_{g \in \mathcal{G}_{L^2 \|b - b^*\|_{\mathcal{B}}^2}} \|g - \mathcal{T}(b - b^*)\|_2 \le \eta < \infty.$$

Also, let $g_{\hat{b}} = \arg\inf_{g \in \mathcal{G}_{L^2 \|\hat{b} - b^*\|_{\mathcal{B}}^2}} \|g - \mathcal{T}(\hat{b} - b^*)\|_2$.

When $\|g_{\hat{b}}\|_2 \le \delta$, then

$$\|\mathcal{T}(\hat{b} - b^*)\|_2 \le \|g_{\hat{b}}\| + \|g - \mathcal{T}(b - b^*)\|_2 \le \delta + \eta;$$

if $\|g_{\hat{b}}\|_2 \ge \delta$, let $r = \frac{\delta}{2\|g_{\hat{b}}\|_2} \in [0, \frac{1}{2}]$, and $rg_{\hat{b}} \in \mathcal{G}_{L^2 \|\hat{b} - b^*\|_{\mathcal{B}}^2}$ since $\mathcal{G}$ is star-shaped. Hence,

$$\sup_{g \in \mathcal{G}} \Psi_n(\hat{b}, g) - \Psi_n(b^*, g) - 2\lambda \left( \|g\|_{\mathcal{G}}^2 + \frac{U}{\delta^2}\|g\|_{2,n}^2 \right) \ge \Psi_n(\hat{b}, rg_{\hat{b}}) - \Psi_n(b^*, rg_{\hat{b}}) - 2\lambda \left( \|rg_{\hat{b}}\|_{\mathcal{G}}^2 + \frac{U}{\delta^2}\|rg_{\hat{b}}\|_{2,n}^2 \right)$$

$$= \underbrace{r\left( \Psi_n(\hat{b}, g_{\hat{b}}) - \Psi_n(b^*, g_{\hat{b}}) \right)}_{(i)} - 2\lambda \underbrace{r^2 \left( \|g_{\hat{b}}\|_{\mathcal{G}}^2 + \frac{U}{\delta^2}\|g_{\hat{b}}\|_{2,n}^2 \right)}_{(ii)}.$$

For $(ii)$,

$$r^2 \left( \|g_{\hat{b}}\|_{\mathcal{G}}^2 + \frac{U}{\delta^2}\|g_{\hat{b}}\|_{2,n}^2 \right) \le \frac{1}{4}\|g_{\hat{b}}\|_{\mathcal{G}}^2 + \frac{r^2 U}{\delta^2}\|g_{\hat{b}}\|_{2,n}^2. \tag{15}$$

By Equation (9), with probability at least $1 - \zeta$,

$$\frac{r^2 U}{\delta^2}\|g_{\hat{b}}\|_{2,n}^2 \le \frac{r^2 U}{\delta^2}\left[ \|g_{\hat{b}}\|_2^2 + \big| \|g_{\hat{b}}\|_{2,n}^2 - \|g_{\hat{b}}\|_2^2 \big| \right] \le \frac{r^2 U}{\delta^2}\left( \frac{3}{2}\|g_{\hat{b}}\|_2^2 + \delta^2(1 + \frac{\|g_{\hat{b}}\|_{\mathcal{G}}^2}{3U}) \right).$$

Substitute this into Equation (15) and we get

$$r^2 \left( \|g_{\hat{b}}\|_{\mathcal{G}}^2 + \frac{U}{\delta^2}\|g_{\hat{b}}\|_{2,n}^2 \right) \le \frac{1}{4}\|g_{\hat{b}}\|_{\mathcal{G}}^2 + \frac{r^2 U}{\delta^2}\left( \frac{3}{2}\|g_{\hat{b}}\|_2^2 + \delta^2(1 + \frac{\|g_{\hat{b}}\|_{\mathcal{G}}^2}{3U}) \right)$$

$$\le \left( \frac{1}{4}\|g_{\hat{b}}\|_{\mathcal{G}}^2 + \frac{1}{12}\|g_{\hat{b}}\|_{\mathcal{G}}^2 \right) + \frac{3U r^2}{2\delta^2}\|g_{\hat{b}}\|_2^2 + \frac{1}{4}U \tag{16}$$

$$= \frac{1}{3}\|g_{\hat{b}}\|_{\mathcal{G}}^2 + (\frac{3}{8} + \frac{1}{4})U$$

$$\le \frac{1}{3}L^2\|\hat{b} - b^*\|_{\mathcal{B}}^2 + \frac{5}{8}U.$$

For $(i)$, we consider function class

$$\mathcal{J}_{B,L^2 B} = \left\{ ((s, a, s'), (r, s, a)) \mapsto \alpha(b(s, a, s') - b^*(s, a, s'))g_b^{L^2 B}(r, s, a) \mid b - b^* \in \mathcal{B}_B, \alpha \in [0, 1] \right\},$$

where $g_b^{L^2 B}(r, s, a) = \arg\inf_{g \in \mathcal{G}_{L^2 B}} \|g - \mathcal{T}(b - b^*)\|_2$. Choose $\delta = \delta_n^{\mathcal{J}} + c_0\sqrt{\frac{\log(c_1/\zeta)}{n}}$, where $\delta_n^{\mathcal{J}}$ is the upper bound of the empirical critical radii of function class $\mathcal{J}_{B,L^2 B}$. Choose loss function $\mathcal{L} = (b - b^*)f$, then by Lemma H.3, we have with probability at least $1 - \zeta$, for all $b \in \mathcal{B}$ and $g \in \mathcal{G}$,

$$|(\Psi_n(b, g_b) - \Psi_n(b^*, g_b)) - (\Psi(b, g_b) - \Psi(b^*, g_b))| \le 18\delta(\|(b^* - b)g_b\|_2 + \delta)$$

$$\le 18\delta(\|g_b\|_2 + \delta),$$

since $b - b^* \in \mathcal{B}_B$, which is 1-uniformly bounded.

When $\|b - b^*\|_{\mathcal{B}}^2 > B$, we rescale the function by $\frac{\sqrt{B}}{\|b-b^*\|_{\mathcal{B}}}(b - b^*)$, and similarly, we obtain that with probability at least $1 - \zeta$,

$$|(\Psi_n(b, g_b) - \Psi_n(b^*, g_b)) - (\Psi(b, g_b) - \Psi(b^*, g_b))| \leq 18\delta(\|g_b\|_2 + \delta) \max\left\{1, \frac{\|b - b^*\|_{\mathcal{B}}^2}{B}\right\}.$$

Therefore, with probability at least $1 - \zeta$, for any $g \in \mathcal{G}$,

$$r\left(\Psi_n(\hat{b}, g_{\hat{b}}) - \Psi_n(b^*, g_{\hat{b}})\right) \geq r\left(\Psi(\hat{b}, g_{\hat{b}}) - \Psi(b^*, g_{\hat{b}})\right) - r|(\Psi_n(b, g_{\hat{b}}) - \Psi_n(b^*, g_{\hat{b}})) - (\Psi(b, g_{\hat{b}}) - \Psi(b^*, g_{\hat{b}}))|$$

$$\geq \underbrace{r\left(\Psi(\hat{b}, g_{\hat{b}}) - \Psi(b^*, g_{\hat{b}})\right)}_{(A)} - \underbrace{18\delta r(\|g_{\hat{b}}\|_2 + \delta) \max\left\{1, \frac{\|\hat{b} - b^*\|_{\mathcal{B}}^2}{B}\right\}}_{(B)}.$$

For $(A)$,

$$r\left(\Psi(\hat{b}, g_{\hat{b}}) - \Psi(b^*, g_{\hat{b}})\right) = \frac{\delta}{2\|g_{\hat{b}}\|_2} \mathbb{E}\left[(\hat{b}(S, A, S') - b^*(S, A, S'))g_{\hat{b}}(R, S, A) \,|\, O_t = 1\right]$$

$$= \frac{\delta}{2\|g_{\hat{b}}\|_2} \mathbb{E}\left[g_{\hat{b}}(R, S, A)\mathbb{E}\left(\hat{b}(S, A, S') - b^*(S, A, S') \mid R, S, A, O_t = 1\right) \,|\, O_t = 1\right]$$

$$= \frac{\delta}{2\|g_{\hat{b}}\|_2} \mathbb{E}\left\{g_{\hat{b}}(R, S, A)\left[\mathcal{T}(\hat{b} - b^*)(R, S, A)\right]\right\}$$

$$= \frac{\delta}{2\|g_{\hat{b}}\|_2} \mathbb{E}\left\{(g_{\hat{b}}(R, S, A))^2 - g_{\hat{b}}(R, S, A)[g_{\hat{b}}(R, S, A) - \mathcal{T}(\hat{b} - b^*)(R, S, A)]\right\}$$

$$= \frac{\delta}{2\|g_{\hat{b}}\|_2} \left\{\|g_{\hat{b}}\|_2^2 - \left\{\mathbb{E}g_{\hat{b}}(R, S, A)[g_{\hat{b}}(R, S, A) - \mathcal{T}(\hat{b} - b^*)(R, S, A)]\right\}\right\}$$

$$\geq \frac{\delta}{2\|g_{\hat{b}}\|_2} \left\{\|g_{\hat{b}}\|_2^2 - \|g_{\hat{b}}\|_2\|g_{\hat{b}} - \mathcal{T}(\hat{b} - b^*))\|_2\right\}$$

$$= \frac{\delta}{2} \left\{\|g_{\hat{b}}\|_2 - \|g_{\hat{b}} - \mathcal{T}(\hat{b} - b^*))\|_2\right\}$$

$$\geq \frac{\delta}{2} \left\{\|\mathcal{T}(\hat{b} - b^*)\|_2 - 2\|g_{\hat{b}} - \mathcal{T}(\hat{b} - b^*))\|_2\right\}$$

$$\geq \frac{\delta}{2} \left\{\|\mathcal{T}(\hat{b} - b^*)\|_2 - 2\eta\right\}.$$

For $(B)$,

$$18\delta r(\|g_{\hat{b}}\|_2 + \delta) \max\left\{1, \frac{\|\hat{b} - b^*\|_{\mathcal{B}}^2}{B}\right\} = 18\delta r(\frac{\delta}{2r} + \delta) \max\left\{1, \frac{\|\hat{b} - b^*\|_{\mathcal{B}}^2}{B}\right\}$$

$$= (9\delta^2 + 18r\delta^2) \max\left\{1, \frac{\|\hat{b} - b^*\|_{\mathcal{B}}^2}{B}\right\}$$

$$\leq 18\delta^2 + \frac{18\delta^2\|\hat{b} - b^*\|_{\mathcal{B}}^2}{B}.$$

So when $\|g_{\hat{b}}\|_2 \geq \delta$, with probability at least $1 - 2\zeta$,

$$(i) = r\left(\Psi_n(\hat{b}, g_{\hat{b}}) - \Psi_n(b^*, g_{\hat{b}})\right) \geq (A) - (B)$$

$$\geq \frac{\delta}{2} \left\{\|\mathcal{T}(\hat{b} - b^*)\|_2 - 2\eta\right\} - 18\delta^2 - \frac{18\delta^2\|\hat{b} - b^*\|_{\mathcal{B}}^2}{B}.$$

Therefore the lower bound of $\sup_{g\in\mathcal{G}} \Psi_n(\hat{b},g) - \Psi_n(b^*,g) - 2\lambda\left(\|g\|_{\mathcal{G}}^2 + \frac{U}{\delta^2}\|g\|_{2,n}^2\right)$ is given by

$$\sup_{g\in\mathcal{G}} \Psi_n(\hat{b},g) - \Psi_n(b^*,g) - 2\lambda\left(\|g\|_{\mathcal{G}}^2 + \frac{U}{\delta^2}\|g\|_{2,n}^2\right)$$

$$\geq(i) - 2\lambda(ii)$$

$$\geq \frac{\delta}{2}\left\{\|\mathcal{T}(\hat{b}-b^*)\|_2 - 2\eta\right\} - 18\delta^2 - \frac{18\delta^2\|\hat{b}-b^*\|_{\mathcal{B}}^2}{B} - 2\lambda\left(\frac{1}{3}L^2\|\hat{b}-b^*\|_{\mathcal{B}}^2 + \frac{5}{8}U\right)$$

$$\geq \frac{\delta}{2}\|\mathcal{T}(\hat{b}-b^*)\|_2 - \eta\delta - 18\delta^2 - \left(\frac{18\delta^2}{B} + \frac{2\lambda L^2}{3}\right)\|\hat{b}-b^*\|_{\mathcal{B}}^2 - \frac{5}{4}\lambda U.$$

### E.3. Combining the upper and lower bounds

Combine the upper bound and lower bound, and then we have either $\|g_{\hat{b}}\|_2 \leq \delta$, or with probability at least $1-3\zeta$, for all $b\in\mathcal{B}$,

$$\frac{\delta}{2}\|\mathcal{T}(\hat{b}-b^*)\|_2 - \eta\delta - 18\delta^2 - \left(\frac{18\delta^2}{B} + \frac{2\lambda L^2}{3}\right)\|\hat{b}-b^*\|_{\mathcal{B}}^2 - \frac{5}{4}\lambda U \leq 2\lambda U + \left(72 + \frac{10\times 18^2}{C_1}\right)\delta^2 + \lambda\mu\left(\|b^*\|_{\mathcal{B}}^2 - \|\hat{b}\|_{\mathcal{B}}^2\right).$$

So

$$\frac{\delta}{2}\|\mathcal{T}(\hat{b}-b^*)\|_2 \leq \frac{13}{4}\lambda U + \eta\delta + \lambda\mu\left(\|b^*\|_{\mathcal{B}}^2 - \|\hat{b}\|_{\mathcal{B}}^2\right) + \left(90 + \frac{10\times 18^2}{C_1}\right)\delta^2 + \left(\frac{18\delta^2}{B} + \frac{2\lambda L^2}{3}\right)\|\hat{b}-b^*\|_{\mathcal{B}}^2$$

$$\leq \frac{13}{4}\lambda U + \eta\delta + \lambda\mu\left(\|b^*\|_{\mathcal{B}}^2 - \|\hat{b}\|_{\mathcal{B}}^2\right) + \left(90 + \frac{10\times 18^2}{C_1}\right)\delta^2 + 2\lambda\left(\frac{18\delta^2}{\lambda B} + \frac{2L^2}{3}\right)\left(\|b^*\|_{\mathcal{B}}^2 + \|\hat{b}\|_{\mathcal{B}}^2\right).$$

If $\mu \geq \frac{36\delta^2}{\lambda B} + \frac{4L^2}{3}$, then

$$\|\mathcal{T}(\hat{b}-b^*)\|_2 \leq \frac{2}{\delta}\left(\frac{13}{4}\lambda U + \eta\delta + \lambda\mu\left(\|b^*\|_{\mathcal{B}}^2 - \|\hat{b}\|_{\mathcal{B}}^2\right) + \left(90 + \frac{10\times 18^2}{C_1}\right)\delta^2 + \lambda\mu\left(\|b^*\|_{\mathcal{B}}^2 + \|\hat{b}\|_{\mathcal{B}}^2\right)\right)$$

$$= \frac{2}{\delta}\left(\frac{13}{4}\lambda U + \eta\delta + 2\lambda\mu\|b^*\|_{\mathcal{B}}^2 + \left(90 + \frac{10\times 18^2}{C_1}\right)\delta^2\right)$$

$$\leq \frac{13}{2}\frac{\lambda U}{\delta} + 2\eta + 4\frac{\lambda\mu}{\delta}\|b^*\|_{\mathcal{B}}^2 + \left(180 + \frac{20\times 18^2}{C_1}\right)\delta.$$

Suppose $\lambda \leq \frac{C_2\delta^2}{U}$, then with probability at least $1-4\zeta$,

$$\|\mathcal{T}(\hat{b}-b^*)\|_2 \leq \frac{13}{2}\frac{\lambda U}{\delta} + 2\eta + 4\frac{\lambda\mu}{\delta}\|b^*\|_{\mathcal{B}}^2 + \left(180 + \frac{20\times 18^2}{C_1}\right)\delta$$

$$\leq \frac{13}{2}C_2\delta + 2\eta + 4C_2\mu\delta\|b^*\|_{\mathcal{B}}^2 + \left(180 + \frac{20\times 18^2}{C_1}\right)\delta$$

$$\leq 4C_2\mu\delta\|b^*\|_{\mathcal{B}}^2 + \left(\frac{13}{2}C_2 + 180 + \frac{20\times 18^2}{C_1}\right)\delta + 2\eta$$

$$\lesssim \delta\max\left\{1, \|b^*\|_{\mathcal{B}}^2\right\}.$$

## F. Proof of Corollary 5.6

Recall that the product class is denoted by

$$\mathcal{J}_{B,L^2 B}^{(t)} := \left\{((s,a,s'),(r,s,a)) \mapsto \alpha(b_t(s,a,s') - b_t^*(s,a,s'))g_{b,t}^{L^2 B}(r,s,a) \mid b_t - b_t^* \in \mathcal{B}_B^{(t)}, \alpha\in[0,1]\right\}.$$

Define the tensor product of two RKHSs $\mathcal{B}^{(t)}$ and $\mathcal{G}^{(t)}$ as $\mathcal{H}_\otimes^{(t)}$ endowed with kernel $K_{\otimes,t}\left((x,y),(x',y')\right) := K_{B,t}(x,x')\,K_{G,t}(y,y')$. Then, one can verify that $\mathcal{J}_{B,L^2 B}^{(t)}$ satisfies

$$\mathcal{J}_{B,L^2 B}^{(t)} \subseteq \left\{f\in\mathcal{H}_\otimes^{(t)} : \|f\|_{\mathcal{H}_\otimes^{(t)}} \leq \sqrt{B_t}\sqrt{L^2 B_t}\right\} =: \mathcal{H}_{\otimes,LB}^{(t)}.$$

By Lemma H.5, we have that

$$\mathcal{R}_{n_t}(\mathcal{J}_{B,L^2B}^{(t)}, \delta) \leq LB\sqrt{\frac{2}{n_t}}\sqrt{\sum_{i=1}^{\infty}\sum_{j=1}^{\infty}\min\{\mu_{t,i}^{\mathcal{B}}\mu_{t,j}^{\mathcal{G}}, \delta^2\}}. \tag{17}$$

Under the polynomial eigen decay assumptions $\mu_{t,i}^{\mathcal{B}} \lesssim i^{-2\alpha_B}$ and $\mu_{t,j}^{\mathcal{G}} \lesssim j^{-2\alpha_G}$ with $\alpha_{\min} := \min\{\alpha_B, \alpha_G\}$, by Lemma H.6, the tensor-product spectrum admits the bound

$$\sum_{i=1}^{\infty}\sum_{j=1}^{\infty}\min\{\mu_{t,i}^{\mathcal{B}}\mu_{t,j}^{\mathcal{G}}, \delta^2\} \lesssim \delta^{2-\frac{1}{\alpha_{\min}}}\log\frac{1}{\delta}.$$

plugging this into Equation (17) yields

$$\delta_{n_t}^{\mathcal{J}} \lesssim LB\, n_t^{-\frac{\alpha_{\min}}{2\alpha_{\min}+1}}\log n_t.$$

Similarly, we have

$$\delta_{n_t}^{\mathcal{G}} \lesssim \sqrt{U_t}\, n_t^{-\frac{\alpha_G}{2\alpha_G+1}}\log n_t.$$

Therefore, $\delta_{n_t} = \max\{\delta_{n_t}^{\mathcal{G}}, \delta_{n_t}^{\mathcal{J}}\}$ satisfies

$$\delta_{n_t} \lesssim \max\{\sqrt{U_t}, LB\}\, n_t^{-\frac{\alpha_{\min}}{2\alpha_{\min}+1}}\log n_t,$$

and the claim follows by Theorem 5.5.

The min-max estimation problem of $b_t$ has a closed-form solution, which is discussed in Dikkala et al. (2020) Appendix E.3.

## G. Proof of Theorem 5.9

### G.1. Error Decomposition

The estimation error bound can be decomposed as

$$|\mathbb{E}[V_1^{\pi}] - \widehat{V}(\pi)| \leq \underbrace{|\mathbb{E}[V_1^{\pi}] - \mathbb{E}_n[V_1^{\pi}]|}_{(I)} + \underbrace{|\mathbb{E}[V_1^{\pi}] - \mathbb{E}[\widehat{V}_1^{\pi}]|}_{(II)} + \underbrace{|\mathbb{E}(V_1^{\pi} - \widehat{V}_1^{\pi}) - \mathbb{E}_n(V_1^{\pi} - \widehat{V}_1^{\pi})|}_{(III)},$$

where $\mathbb{E}_n[V_1^{\pi}] := \frac{1}{n}\sum_{i=1}^{n}V_1^{\pi}(S_{1,i}, 0)$.

### G.2. Bound of $(I)$

For $(I)$, since rewards are bounded in $[-1, 1]$, by Hoeffding inequality, with probability at least $1 - \zeta$,

$$|\mathbb{E}[V_1^{\pi}] - \mathbb{E}_n[V_1^{\pi}]| \lesssim \|V_1^{\pi}\|_{\infty}\sqrt{\frac{\log(c_1/\zeta)}{n}} \leq T\sqrt{\frac{\log(c_1/\zeta)}{n}},$$

where constant $c_1 > 0$.

### G.3. Bound of $(II)$

For $(II)$,

$$\begin{aligned}
|\mathbb{E}[V_1^{\pi}] - \mathbb{E}[\widehat{V}_1^{\pi}]| &\leq \|V_1^{\pi} - \widehat{V}_1^{\pi}\|_2 \\
&= \|\sum_a \pi_1(a \mid S_1, O_0)(Q_1^{\pi}(S_1, a) - \widehat{Q}_1(S_1, a))\|_2 \\
&= \left(\mathbb{E}\left[\left(\sum_a \pi_1(a \mid S_1, O_0 = 0)(Q_1^{\pi}(S_1, a) - \widehat{Q}_1(S_1, a))\right)^2\right]\right)^{1/2} \\
&\leq \left(\mathbb{E}\left[\sum_a \pi_1(a \mid S_1, O_0 = 0)(Q_1^{\pi}(S_1, a) - \widehat{Q}_1(S_1, a))^2\right]\right)^{1/2} \\
&:= \|Q_1^{\pi} - \widehat{Q}_1\|_{2,\pi},
\end{aligned}$$

where $Q_t^\pi(s, a) = \mathbb{E}(R_t + V_{t+1}^\pi \mid s, a)$. Since there is no shift on marginal distributions of states $\tilde{d}_t^\pi$ and $\tilde{d}_t^b$ when $t = 1$, by Assumption 2.3,

$$\|Q_1^\pi - \widehat{Q}_1\|_{2,\pi}^2 = \mathbb{E}_{(S_1,O_0)\sim \tilde{d}_1^\pi}\Big[\sum_a \pi_1(a \mid S_1, O_0 = 0)(Q_1^\pi(S_1, a) - \widehat{Q}_1(S_1, a))^2\Big]$$

$$= \mathbb{E}_{(S_1,O_0)\sim \tilde{d}_1^b}\Big[\sum_a \pi_1(a \mid S_1, O_0 = 0)(Q_1^\pi(S_1, a) - \widehat{Q}_1(S_1, a))^2\Big]$$

$$\leq \kappa_1 \mathbb{E}_{(S_1,O_0)\sim \tilde{d}_1^b}\Big[\sum_a \pi_1^b(a \mid S_1)(Q_1^\pi(S_1, a) - \widehat{Q}_1(S_1, a))^2\Big]$$

$$:= \kappa_1 \|Q_1^\pi - \widehat{Q}_1\|_{2,\pi^b}^2,$$

and for simplicity we write $\|Q_1^\pi - \widehat{Q}_1\|_2^2$ instead of $\|Q_1^\pi - \widehat{Q}_1\|_{2,\pi^b}^2$. $\widehat{Q}_t$ is estimated from penalized nonparametric least square problem Equation (8).

$\|\widehat{Q}_t - Q_t^\pi\|_2$ involves the estimation error of the fitted $Q$-functions produced by FQE. Unlike standard supervised regression, the regression targets in FQE are pseudo-labels that depend on nuisance estimates and on future-stage fitted values. Concretely, at stage $t < T$, the target takes the form

$$y_{t,i} = \widehat{\widetilde{R}}_{t,i} + \widehat{V}_{t+1}^\pi(S_{t+1,i}, O_{t,i}),$$

where $\widehat{\widetilde{R}}_{t,i}$ depends on the estimated bridge $\hat{b}_t$ and $\widehat{V}_{t+1}^\pi$ depends on $\widehat{Q}_{t+1}$. Therefore, the regression noise and the regression function are statistically coupled through the common data, and a direct analysis of $\|\widehat{Q}_t - Q_t^\pi\|_2$ typically leads to non-negligible cross terms that are difficult to control without additional device such as sample splitting or cross-fitting.

To decouple the effect of nuisance estimation from the intrinsic regression error, we introduce an oracle comparator $\widehat{Q}_t^*$, defined as the solution of the same penalized regression problem as $\widehat{Q}_t$ but trained on an oracle pseudo-label $y_t^*$, in which the nuisance components are replaced by their population counterparts.

$$\widehat{Q}_t^* = \arg\min_{f \in \mathcal{Q}^{(t)}} \frac{1}{n} \sum_{i=1}^n \big(f(S_{t,i}, A_{t,i}) - y_{t,i}^*\big)^2 + \lambda_{Q,t}\|f\|_{\mathcal{Q}^{(t)}}^2, \tag{18}$$

where

$$y_{t,i}^* = \begin{cases} \widetilde{R}_{t,i}, & t = T, \\ \widetilde{R}_{t,i} + V_{t+1}^\pi(S_{t+1,i}, O_{t,i}), & t < T, \end{cases}$$

and $V_{t+1}^\pi(S_{t+1,i}, O_{t,i}) = \sum_a \pi_{t+1}(a \mid S_{t+1}, O_t)Q_{t+1}^\pi(S_{t+1}, a)$.

Then we have

$$\|\widehat{Q}_t - Q_t^\pi\|_2 = \|(\widehat{Q}_t - \widehat{Q}_t^*) + (\widehat{Q}_t^* - Q_t^\pi)\|_2$$

$$\leq \underbrace{\|\widehat{Q}_t - \widehat{Q}_t^*\|_2}_{(a)} + \underbrace{\|\widehat{Q}_t^* - Q_t^\pi\|_2}_{(b)}.$$

For $(a)$, since $\widehat{Q}_t, \widehat{Q}_t^*$ are estimated from Equation (8) and *Equation* (18), it can be verified that

$$\|\widehat{Q}_t - \widehat{Q}_t^*\|_{2,n} \leq \|y_t - y_t^*\|_{2,n} \leq \sqrt{2}\|\widehat{\widetilde{R}}_t - \widetilde{R}_t\|_{2,n} + \sqrt{2}\|\widehat{V}_{t+1}^\pi - V_{t+1}^\pi\|_{2,n}.$$

Since

$$\|\widehat{\widetilde{R}}_t - \widetilde{R}_t\|_{2,n} = \|(1 - O_t)(\hat{b}_t - b_t^*)\|_{2,n} \leq \|\hat{b}_t - b_t^*\|_{2,n},$$

and

$$\|\widehat{V}_{t+1}^\pi - V_{t+1}^\pi\|_{2,n} := \Big\| \sum_a \pi_{t+1}(a \mid S_{t+1}, O_t)\big(\widehat{Q}_{t+1}(S_{t+1}, a) - Q_{t+1}^\pi(S_{t+1}, a)\big) \Big\|_{2,n}$$

$$= \Big(\frac{1}{n}\sum_{i=1}^n \big[\sum_a \pi_{t+1}(a \mid S_{t+1,i}, O_{t,i})\big(\widehat{Q}_{t+1}(S_{t+1,i}, a) - Q_{t+1}^\pi(S_{t+1,i}, a)\big)\big]^2\Big)^{1/2}$$

$$\le \Big(\frac{1}{n}\sum_{i=1}^n \sum_a \pi_{t+1}(a \mid S_{t+1,i}, O_{t,i})\big(\widehat{Q}_{t+1}(S_{t+1,i}, a) - Q_{t+1}^\pi(S_{t+1,i}, a)\big)^2\Big)^{1/2}$$

$$:= \|\widehat{Q}_{t+1} - Q_{t+1}^\pi\|_{2,n,\pi},$$

then

$$\|\widehat{Q}_t - \widehat{Q}_t^*\|_{2,n} \le \sqrt{2}\|\hat{b}_t - b_t^*\|_{2,n} + \sqrt{2}\|\widehat{Q}_{t+1} - Q_{t+1}^\pi\|_{2,n,\pi}. \tag{19}$$

Since $\mathcal{Q}^{(t)}$ is $(T - t + 1)$-uniformly bounded, we consider scaling $\mathcal{Q}^{(t)}$ by $(T - t + 1)$ for convenience.

According to Fischer & Steinwart (2020), under mild conditions, the RKHS norm of kernel ridge regression estimators is bounded with high probability. and construct RKHS ball $\mathcal{Q}_{R_Q}^{(t)} = \{Q \in \mathcal{Q}^{(t)} : \|Q\|_{\mathcal{Q}^{(t)}} \le R_Q\}$ for all $t = 1, \dots, T$. Denote difference class $\Delta \mathcal{Q}^{(t)} := \{\Delta_Q \mid \Delta_Q = Q_1 - Q_2, \quad Q_1, Q_2 \in \mathcal{Q}_{R_Q}^{(t)}\}$. Let $\bar{\delta}_{\Delta_Q^{(t)},n}$ be the upper bound of the empirical critical radii of scaled function class $\Delta \mathcal{Q}^{(t)}$.

For RKHS $\mathcal{B}^{(t)}$, Proposition 9 in Dikkala et al. (2020) gives the closed form of the inner maximization, which implies that $\|\hat{b}_t\|_{\mathcal{B}^{(t)}}$ can be bounded by a constant $R_B$. Thus, consider RKHS ball $\mathcal{B}_{R_B}^{(t)} = \{b_t \in \mathcal{B}^{(t)}, \|b_t\|_{\mathcal{B}^{(t)}} \le R_B\}$. Define difference class $\Delta \mathcal{B}^{(t)} = \{\Delta_b = b_1 - b_2, \quad b_1, b_2 \in \mathcal{B}_{R_B}^{(t)}\}$ and let $\bar{\delta}_{\Delta_b^{(t)},n}$ be the upper bound of the empirical critical radii of $\Delta \mathcal{B}^{(t)}$. Then, we define $\bar{\delta}_{\Delta_t} = \max\{\bar{\delta}_{\Delta_Q^{(t)},n}, \bar{\delta}_{\Delta_Q^{(t+1)},n}, \bar{\delta}_{\Delta_b^{(t)},n}\}$ where $\delta_{\Delta_t} = \bar{\delta}_{\Delta_t} + c_0\sqrt{\frac{\log(c_1/\zeta)}{n}}$ for some $c_0, c_1 > 0$. Let $\|\widehat{Q}_{t+1} - Q_{t+1}^\pi\|_{2,b,\pi}^2 := \mathbb{E}_{(S_{t+1}, O_t) \sim \tilde{d}_{t+1}^b}\Big[\sum_{a \in \mathcal{A}} \pi_{t+1}(a \mid S_{t+1}, O_t)\big(\widehat{Q}_{t+1}(S_{t+1}, a) - Q_{t+1}^\pi(S_{t+1}, a)\big)^2\Big]$.

Applying Lemma H.2 on both sides of Equation (19), we have with probability at least $1 - \zeta$,

$$\|\widehat{Q}_t - \widehat{Q}_t^*\|_2 \lesssim \|\hat{b}_t - b_t^*\|_2 + \|\widehat{Q}_{t+1} - Q_{t+1}^\pi\|_{2,b,\pi} + (T - t + 1)\delta_{\Delta_t}$$

$$(\text{Assumption 2.3 (2)}) \lesssim \|\hat{b}_t - b_t^*\|_2 + \sqrt{\kappa_{t+1}}\|\widehat{Q}_{t+1} - Q_{t+1}^\pi\|_2 + (T - t + 1)\delta_{\Delta_t}. \tag{20}$$

$(b)$ corresponds to a standard penalized least square estimation error. Since $\|Q_t^*\|_{\mathcal{Q}^{(t)}}$ is bounded, by Lemma H.7, with probability at least $1 - \zeta$, $(b)$ is bounded by

$$\|\widehat{Q}_t^* - Q_t^\pi\|_2 \lesssim (\delta_{\Delta_Q^{(t)}} + \sqrt{\lambda_{Q,t}})(T - t + 1),$$

where $\delta_{\Delta_Q^{(t)}} = \delta_{\Delta_Q^{(t)},n} + c_0\sqrt{\frac{\log(c_1 T/\zeta)}{n}}$ for some $c_0, c_1 > 0$.

Therefore, with probability at least $1 - \zeta/T$,

$$\|\widehat{Q}_t - Q_t^\pi\|_2 \le \|\widehat{Q}_t - \widehat{Q}_t^*\|_2 + \|\widehat{Q}_t^* - Q_t^\pi\|_2$$

$$\lesssim \|\hat{b}_t - b_t^*\|_2 + \sqrt{\kappa_{t+1}}\|\widehat{Q}_{t+1} - Q_{t+1}^\pi\|_2 + (T - t + 1)\delta_{\Delta_t} + (\delta_{\Delta_Q^{(t)}} + \sqrt{\lambda_{Q,t}})(T - t + 1).$$

Applying backward induction from $t = T$ down to $t = 1$ yields the bound for $(II)$ with probability at least $1 - \zeta$:

$$
\begin{aligned}
|\mathbb{E}[V_1^\pi] - \mathbb{E}[\widehat{V}_1^\pi]| &\leq \|V_1^\pi - \widehat{V}_1^\pi\|_2 \\
&\leq \sqrt{\kappa_1} \|\widehat{Q}_1 - Q_1^\pi\|_2 \\
&\lesssim \sum_{t=1}^T \Big( \prod_{j=1}^T \sqrt{\kappa_j} \Big) \Big[ \|\hat{b}_t - b_t^*\|_2 + (T - t + 1)\delta_{\Delta_t} + (\delta_{\Delta_Q^{(t)}} + \sqrt{\lambda_{Q,t}})(T - t + 1) \Big] \\
&\lesssim K \sum_{t=1}^T \Big[ \tau_t \delta_t (1 + \|b_t^*\|_{\mathcal{B}^{(t)}}^2) + (\delta_{\Delta_t} + \delta_{\Delta_Q^{(t)}} + \sqrt{\lambda_{Q,t}})(T - t + 1) \Big].
\end{aligned}
$$

### G.4. Bound of $(III)$

For $(III)$, we first define function class $\Delta \mathcal{V}^{(t)} = \{ \Delta = V_1 - V_2 \mid V_1, V_2 \in \mathcal{V}_{R_V}^{(t)} \}$, where $\mathcal{V}_{R_V}^{(t)}$ is a $(T - t + 1)$-uniformly bounded function class of value functions at time $t$, induced from $\mathcal{Q}^{(t)}$ under operator $\Pi_t$: $\mathcal{V}_{R_V}^{(t)} = \{\Pi_t Q : Q \in \mathcal{Q}_{R_Q}^{(t)}\}$. Here linear operator $\Pi_t$ is defined as $(\Pi_t Q)(s, o-) = \sum \pi_t(a \mid s, o_-) Q(s, a)$. We choose the cost function as $\mathcal{L}(f(X), Y) = f(X)$ and apply Lemma H.1. We here also scale $\mathcal{V}_{R_V}^{(t)}$ by $(T - t + 1)$.

Then, with probability at least $1 - \zeta$,

$$
|\mathbb{E}(V_1^\pi - \widehat{V}_1^\pi) - \mathbb{E}_n(V_1^\pi - \widehat{V}_1^\pi)| \lesssim \delta_{\Delta_V^{(1)}}(\|V_1^\pi - \widehat{V}_1^\pi\|_2 + T\delta_{\Delta_V^{(1)}}),
$$

where $\delta_{\Delta_V^{(1)}} = \bar{\delta}_{\Delta_V^{(1)}, n} + c_0 \sqrt{\frac{\log(c_1/\zeta)}{n}}$, and $\bar{\delta}_{\Delta_V^{(1)}, n}$ is the upper bound of the empirical critical radii of scaled function class $\Delta \mathcal{V}^{(1)}$. Moreover, $\delta_{\Delta_V^{(1)}} \leq \delta_{\Delta_Q^{(1)}}$.

### G.5. Policy value error bound

Combine the above inequalities, we obtain the policy value estimation error bound with probability at least $1 - \zeta$:

$$
\begin{aligned}
\big| \widehat{V}(\pi) - V(\pi) \big| &\leq (I) + (II) + (III) \\
&\lesssim T\sqrt{\frac{\log(c_1 T/\zeta)}{n}} + (\delta_{\Delta_V^{(1)}} + 1)\|V_1^\pi - \widehat{V}_1^\pi\|_2 + T(\delta_{\Delta_V^{(1)}})^2 \\
&\lesssim T\sqrt{\frac{\log(c_1 T/\zeta)}{n}} + T(\delta_{\Delta_V^{(1)}})^2 + K(\delta_{\Delta_V^{(1)}} + 1)\Big[ \tau_t \delta_t (1 + \|b_t^*\|_{\mathcal{B}^{(t)}}^2) + \\
&\quad (\delta_{\Delta_t} + \delta_{\Delta_Q^{(t)}} + \sqrt{\lambda_{Q,t}})(T - t + 1) \Big] \\
&\lesssim T\sqrt{\frac{\log(c_1 T/\zeta)}{n}} + T(\delta_{\Delta_V^{(1)}})^2 + K(\delta_{\Delta_V^{(1)}} + 1)\Big[ \tau_t \delta_t (1 + \|b_t^*\|_{\mathcal{B}^{(t)}}^2) + \\
&\quad (T - t + 1)\delta_{t,*} \Big] \\
&\lesssim T\sqrt{\frac{\log(c_1 T/\zeta)}{n}} + K\tau_{\max} T \sum_{t=1}^T \delta_{t,*},
\end{aligned}
$$

where $\delta_{t,*}$ as the maximum of the critical radii of difference classes $\Delta \mathcal{Q}^{(t)}$, $\Delta \mathcal{Q}^{(t+1)}$, $\Delta \mathcal{B}^{(t)}$, and $\mathcal{G}_U^{(t)}$ for $t = 1, \ldots, T$, namely $\delta_{\Delta_Q^{(t)}}$, $\delta_{\Delta_Q^{(t+1)}}$, $\delta_{\Delta_B^{(t)}}$ and $\delta_{G^{(t)}}$.

With polynomial decay

$$
\mu_{t,j}^{\mathcal{Q}} \lesssim j^{-2\alpha_Q}, \quad \mu_{t,j}^{\mathcal{B}} \lesssim j^{-2\alpha_B}, \quad \mu_{t,j}^{\mathcal{G}} \lesssim j^{-2\alpha_G}, \quad \alpha_Q, \alpha_B, \alpha_G > 1/2,
$$

the corresponding critical radii satisfy

$$\delta_{\Delta_Q^{(t)}}, \delta_{\Delta_Q^{(t+1)}} \lesssim R_Q^{\frac{1}{2\alpha_Q+1}} n^{-\frac{\alpha_Q}{2\alpha_Q+1}} \log n,$$

$$\delta_{\Delta_B^{(t)}} \lesssim R_B^{\frac{1}{2\alpha_B+1}} n^{-\frac{\alpha_B}{2\alpha_B+1}} \log n,$$

$$\delta_{G^{(t)}} \lesssim U_t^{\frac{1}{2\alpha_G+1}} n^{-\frac{\alpha_G}{2\alpha_G+1}} \log n.$$

Thus, the critical radius $\delta_{t,*}$ satisfies

$$\delta_{t,*} \lesssim \max\{\sqrt{R_Q}, \sqrt{R_B}, \sqrt{U_t}\} n^{-\frac{\alpha_{\min}}{2\alpha_{\min}+1}} \log n,$$

where $\alpha_{\min} = \min\{\alpha_Q, \alpha_B, \alpha_G\}$. Therefore, with probability at least $1 - \zeta$, the policy value is bound by

$$\left|\widehat{V}(\pi) - V(\pi)\right| \lesssim K\tau_{\max}T^2\sqrt{\log(c_1 T/\zeta)} n^{-\frac{\alpha_{\min}}{2\alpha_{\min}+1}} \log n.$$

## H. Auxiliary lemmas

**Lemma H.1** (Wainwright (2019), Theorem 14.20). *Suppose function class $\mathcal{F}$ is symmetric, 1-uniformly bounded, and star-shaped around $f^*$. Let $\delta_n^2 \geq \frac{c}{n}$ be any solution to the inequality $\mathcal{R}_n(\mathcal{F}^*, \delta) \leq \delta^2$, where $\mathcal{F}^* = \{f - f^* \mid f \in \mathcal{F}\}$. Suppose the cost function $\mathcal{L}(f(X), Y)$ is L-Lipschitz in its first argument $f(X)$. Then for all $f \in \mathcal{F}$, with probability at least $1 - c_1 e^{-c_2 n\delta_n^2}$, we have*

$$|\mathbb{E}_n\big(\mathcal{L}(f(x), y) - \mathcal{L}(f^*(x), y)\big) - \mathbb{E}\big(\mathcal{L}(f(x), y) - \mathcal{L}(f^*(x), y)\big)| \leq 10L\delta_n\big(\|f - f^*\|_2 + \delta_n\big).$$

**Lemma H.2** (Wainwright (2019), Theorem 14.1). *Given a star-shaped and b-uniformly bounded function class $\mathcal{F}$, set $\delta_n > 0$ be any solution to $\mathcal{R}(\mathcal{F}, \delta) \leq \frac{\delta^2}{b}$. Then for any $t \geq \delta_n$, with probability at least $1 - c_1 \exp(-c_2 \frac{nt^2}{b^2})$, we have*

$$\left|\|f\|_{2,n}^2 - \|f\|_2^2\right| \leq \frac{1}{2}\|f\|_2^2 + \frac{1}{2}t^2$$

*for all $f \in \mathcal{F}$.*

**Lemma H.3** (Foster & Syrgkanis (2023), Lemma 14). *Consider a 1-uniformly bounded and star-shaped function class $\mathcal{F}$, and pick any $f^* \in \mathcal{F}$. Let $\delta_n^2 \geq c_1 \frac{\log(\log n)}{n}$ be any solution to the inequalities $\mathcal{R}_n(\mathcal{F}_t^*, \delta) \leq \delta^2$ for all $t \in \{1, \ldots, d\}$, where $\mathcal{F}_t^* = \{f_t - f_t^* \mid f_t \in \mathcal{F}|_t\}$. Assume $\mathcal{L}_f$ is L-Lipschitz in its first argument $f$ with respect to its $\ell_2$ norm. Then for all $f \in \mathcal{F}$, for some universal constants $c_2, c_3 > 0$, with probability at least $1 - c_2 e^{-c_3 n\delta_n^2}$, we have*

$$|\mathbb{E}_n\big(\mathcal{L}_f - \mathcal{L}_{f^*}\big) - \mathbb{E}\big(\mathcal{L}_f - \mathcal{L}_{f^*}\big)| \leq 18Ld\delta_n\big(\|f - f^*\|_2 + \delta_n\big).$$

*The outcome $\hat{f}$ of constrained ERM satisfies that with the same probability,*

$$\mathbb{E}_n\big(\mathcal{L}_{\hat{f}} - \mathcal{L}_{f^*}\big) \leq 18Ld\delta_n\big(\|\hat{f} - f^*\|_2 + \delta_n\big).$$

**Lemma H.4** (Wainwright (2019), Example 3.5). *Let $\varepsilon = (\varepsilon_1, \ldots, \varepsilon_n)$ be i.i.d. Rademacher variables taking values in $\{-1, +1\}$ with equal probability. Let $\mathcal{A} \subset \mathbb{R}^n$ be any (possibly infinite) bounded set, and define*

$$Z(\mathcal{A}) := \sup_{a \in \mathcal{A}} \langle a, \varepsilon \rangle = \sup_{a \in \mathcal{A}} \sum_{k=1}^{n} a_k \varepsilon_k.$$

*Let $W(\mathcal{A}) := \sup_{a \in \mathcal{A}} \|a\|_2$. Then for all $t > 0$,*

$$\mathbb{P}\Big(Z(\mathcal{A}) \geq \mathbb{E}[Z(\mathcal{A})] + t\Big) \leq \exp\Big(-\frac{t^2}{16\,W(\mathcal{A})^2}\Big).$$

*Moreover, since $-Z(\mathcal{A}) = \inf_{a \in \mathcal{A}} \langle a, \varepsilon \rangle$ and the same argument applies,*

$$\mathbb{P}\Big(|Z(\mathcal{A}) - \mathbb{E}[Z(\mathcal{A})]| \geq t\Big) \leq 2\exp\Big(-\frac{t^2}{16\,W(\mathcal{A})^2}\Big).$$

**Lemma H.5** (Wainwright (2019), Corollary 14.5). *Let $\mathcal{H}$ be an RKHS with reproducing kernel $K$ and let $\mathcal{F} := \{f \in \mathcal{H} : \|f\|_{\mathcal{H}} \leq 1\}$ be the unit ball. Let $\{\mu_j\}_{j=1}^{\infty}$ denote the non-increasing eigenvalues. Then the local Rademacher complexity satisfies, for any $\delta > 0$,*

$$\mathcal{R}_n(\mathcal{F}, \delta) \leq \sqrt{\frac{2}{n}} \left( \sum_{j=1}^{\infty} \min\{\mu_j, \delta^2\} \right)^{1/2}.$$

*Moreover, let $\{\hat{\mu}_j\}_{j=1}^{n}$ denote the eigenvalues of the renormalized kernel matrix $\mathbf{K} \in \mathbb{R}^{n \times n}$ with entries $\mathbf{K}_{ij} = K(x_i, x_j)/n$. Then the local empirical Rademacher complexity satisfies, for any $\delta > 0$,*

$$\widehat{\mathcal{R}}_n(\mathcal{F}, \delta) \leq \sqrt{\frac{2}{n}} \left( \sum_{j=1}^{n} \min\{\hat{\mu}_j, \delta^2\} \right)^{1/2}.$$

**Lemma H.6** (Krieg (2018), Theorem 1(i)). *Let $\sigma : \mathbb{N} \to \mathbb{R}_+$ be a non-increasing sequence with $\sigma(n) \to 0$. For $d \in \mathbb{N}$, define its $d$-th tensor power*

$$\sigma_d(n_1, \ldots, n_d) = \prod_{k=1}^{d} \sigma(n_k), \quad (n_1, \ldots, n_d) \in \mathbb{N}^d,$$

*and let $\tau : \mathbb{N} \to \mathbb{R}_+$ be the non-increasing rearrangement of $\{\sigma_d(n_1, \ldots, n_d)\}_{(n_1, \ldots, n_d) \in \mathbb{N}^d}$. If for some $s > 0$ one has $\sigma(n) \lesssim n^{-s}$, then*

$$\tau(n) \lesssim n^{-s} (\log n)^{s(d-1)}.$$

**Lemma H.7** (Rademacher analogue of Wainwright (2019), Theorem 13.17). *Let $(x_i, y_i)_{i=1}^{n}$ be i.i.d. with $y_i = f^*(x_i) + \xi_i$, where $\mathbb{E}[\xi_i \mid x_i] = 0$ and $\xi_i$ is conditionally $\sigma$-sub-Gaussian. Let $\mathcal{F}$ be a symmetric, star-shaped class equipped with a Hilbert norm $\|\cdot\|_{\mathcal{F}}$. Consider the penalized least squares estimator*

$$\hat{f} \in \arg\min_{f \in \mathcal{F}} \left\{ \frac{1}{2n} \sum_{i=1}^{n} (y_i - f(x_i))^2 + \lambda_n \|f\|_{\mathcal{F}}^2 \right\}.$$

*Suppose $f^* \in \mathcal{F}$ and $\|f^*\|_{\mathcal{F}} \leq R$. Define the localized difference class*

$$\mathcal{G} := \{\Delta = f - f^* : f \in \mathcal{F}, \|f\|_{\mathcal{F}} \leq R\}.$$

*and the local empirical Rademacher complexity*

$$\widehat{\mathcal{R}}_n(\mathcal{G}, \delta) := \mathbb{E}_{\varepsilon} \left[ \sup_{\Delta \in \mathcal{G} : \|\Delta\|_{2,n} \leq \delta} \left| \frac{1}{n} \sum_{i=1}^{n} \varepsilon_i \Delta(x_i) \right| \, \Big| \, x_{1:n} \right].$$

*Let $\bar{\delta}_n$ be the upper bound of the critical radii satisfying $\widehat{\mathcal{R}}_n(\mathcal{G}, \bar{\delta}_n) \leq \frac{\bar{\delta}_n^2}{32\sigma}$, and define $\delta_n = \bar{\delta}_n + c_0 \sigma \sqrt{\frac{\log(c_1/\zeta)}{n}}$ for some numerical constants $c_0, c_1 > 0$. Assume that $\lambda_n \geq \frac{3}{4}\delta_n^2$, then there exist constant $C_1 > 0$ such that, with probability at least $1 - \zeta$,*

$$\|\hat{f} - f^*\|_2^2 \leq C_1 R^2 (\delta_n^2 + \lambda_n).$$

