# OpenReview forum: "Off-Policy Evaluation for Missingness-Aware Policies in MDPs with Rewards Missing Not at Random"
_ICML.cc/2026/Conference — ICML 2026 regular_

### Official Review · Reviewer_8YRm · 2026-03-05

**Soundness:** 3
**Presentation:** 3
**Significance:** 3
**Originality:** 2
**Overall Recommendation:** 4
**Confidence:** 5

**Summary:**

This paper investigate the problem of Off Policy Evaluation when the rewards aren't missing at random. The authors use FQE for this, and design bridge functions which they learn in a minimax manner. In order to do this, they first show under what conditions such a setup is even identifiable. The main contributions lie in the Proximal-FQE algorithm, which uses these bridge functions, and the theoretical guarantees involve error-bound estimations (Based on Radamacher complexity) and measure of ill-posedness.

**Compliance With Llm Reviewing Policy:**

Affirmed.

**Final Justification:**

My questions have been reasonably discussed and answered. The authors have conducted additional experiments on MIMIC-III and pointed out the difference w.r.t. reward shaping. These were the 2 major drawbacks in the original manuscript which have been addressed, and hence, I recommend acceptance.

**Key Questions For Authors:**

Kindly look at Weaknesses.

**Limitations:**

Limited discussion on limitations, the main limitations are highlighted by me in weaknesses.

**Strengths And Weaknesses:**

Strengths:
1. The problem being addressed is a very practical one, often frequently encountered in POMDP settings in healthcare and other reward-sparse settings. The authors have done a good job formalising the theoretical aspects, following the structure of many standard OPE papers such as (Kuang et al. OPE for Confounded POMDPs https://openreview.net/pdf?id=t3LTjkXDJA). The bounds are correct and provide intuition on how to apply FQE under missing rewards.
2. The paper is well written and easy to follow.

Weaknesses:
1. The paper is missing decent experimental setups, which are important for the paper as the motivation of the problem is a practical one and not a theoretical one. The experimental setup considered is a toy one with no practical significance. Experiments on real-world medical examples from MIMIC-IV or finance is necessary to ground the theoretical results of the paper.
2. A major chunk of related work on Reward Shaping is missing. Reward Shaping [Ng. et al. 1999] is typically used to give feedbacks in the middle of the trajectory in sparse reward scenarios to assist with faster optimisations. Since this paper directly works on missing rewards, comparisons against shaped reward OPE is critical. As an example: See Parbhoo et al. https://offline-rl-neurips.github.io/pdf/56.pdf
3. Are you augmenting the state with all reward missingness indicators of the past? If yes, that also invites a curse-of-horizon discussion on the number of indicators (this scales with |H|) since you are talking about a reward at every step, and there is no Markovian-ness over rewards.
4. The paper only considers FQE. There is no discussion on some recent POMDP-based OPE estimators like Future value functions (Zhang et al. On the curses of future and history in future-dependent value functions for off-policy evaluation. ), Model-based methods (Zhang et al. Statistical Tractability of Off-policy Evaluation of History-dependent Policies in POMDPs), or even representation methods (Majumdar et al. Concept-based Off Policy Evaluation), Kuang et al. OPE for Confounded POMDPs https://openreview.net/pdf?id=t3LTjkXDJA. (This is a somewhat minor weakness, as I would expect the theoretical bounds to hold albeit a different formulation, but it needs to be addressed in the related work )

---

> ### Author Rebuttal · Authors · 2026-03-30
>
> We sincerely thank the reviewer for the detailed and thorough review. We have carefully addressed each concern below.
>
> > W1. Experiments on real-world medical examples from MIMIC-IV or finance is necessary to ground the theoretical results of the paper.
>
> We conduct experiments on real-world MIMIC-III sepsis cohort, following [Raghu et al. (2017)](https://arxiv.org/pdf/1711.09602). The preprocessed dataset contains 13,943 ICU patients, 48-dim states, 25 discrete actions (5 Vasopressor levels × 5 IV fluid levels), and horizon T=10. The reward is the decrease in SOFA $$R_t=-(SOFA_{t+1}-SOFA_t).$$ We impose synthetic MNAR missingness on observed rewards. The observation indicator $$O_t\sim Bernoulli(\sigma(c_0+2R_t^{norm}+0.3S_t^{signal}-0.1A_t^{norm})),$$
> where
> - $R_t^{norm}$ and $A_t^{norm}$ are normalized rewards and actions
> - $S_t^{signal}=\text{normalized Arterial lactate}-0.5*\text{normalized MeanBP}$
> - The intercept $c_0$ is calibrated to target missing rates from 0.2 to 0.8.
>
> We first perform DDQN on the original data. We construct a target conservative policy: when $O_{t-1}=0$, both vasopressor and IV fluid dose levels in $A_t$ are reduced by 1 (clipped at 0). This creates action distributions that differ across missingness-aware states. We split the data by patient ID: 60% for training and 40% for evaluating the policy value.
>
> We compare each method with OracleFQE (uses fully observed rewards), and report absolute error $|\hat V(\pi)-V_{Oracle}|$. We also add two new baselines. ImputeFQE fits a regressor on observed rewards using $(S_t,A_t)$, imputes missing rewards, then applies standard FQE. SCOPE will be introduced later.
>
> See results in https://ibb.co/P3HWd1c. ProxFQE achieves the lowest error across all missing rates.
>
> > W2. ...related work on Reward Shaping is missing... comparisons against shaped reward OPE is critical...
>
> We added the following into related works. Potential-based reward shaping augments the reward by a potential difference to densify sparse feedback while preserving the optimal policy ([Ng et al. 1999](https://dl.acm.org/doi/10.5555/645528.657613)). Later works extend this to dynamic shaping and off-policy learning settings ([Devlin & Kudenko, 2012](https://www.ifaamas.org/Proceedings/aamas2012/papers/2C_3.pdf); [Harutyunyan et al., 2015](https://ai.vub.ac.be/~tbrys/publications/Harutyunyan2015ALA.pdf); [Parbhoo et al. 2020](https://offline-rl-neurips.github.io/pdf/56.pdf)). These are designed to accelerate policy learning or improve optimization under observed rewards, not to identify policy values when the rewards are MNAR. In contrast, our method is designed to recover the full-data conditional mean reward via a bridge function in the setting where the observation probability depends on the latent reward. In our setting, we are targeting MNAR bias correction, not sparse-reward credit assignment alone.
>
> We also add SCOPE (Parbhoo et al., 2020) as a new baseline, which uses per-step importance sampling with potential-based reward shaping.
> - In simulation, across varying sample sizes, horizon lengths, reward mechanisms, SCOPE performs reasonably at low missing rates, while ProxFQE remains stable.
> - In MIMIC-III experiment, SCOPE produces degenerate estimates due to low support overlap for importance sampling. Fewer than 0.5% of trajectories have nonzero IS weight, making the estimator unreliable.
>
> See the simulation results
> - https://ibb.co/Hft7XM1M MSE vs sample size under different missingness rates;
> - https://ibb.co/xSF2jjL5 MSE vs horizon length under different missingness rates;
> - https://ibb.co/hFwMwB0t MSE vs reward type under different missingness rates.
>
> > W3. Are you augmenting the state with all reward missingness indicators of the past?
>
> No. We only augment the state with the most recent indicator $O_{t-1}$. The augmented state is $\tilde S_t=(S_t,O_{t-1})$, which adds exactly one dimension. Assumption 3.2 ensures that the augmented process is Markov, so the past indicators carry no additional information when the target policy only depends on $\tilde S_t$.
>
> Augmenting the state with all past missingness can be a great direction for future works.
>
> > W4. There is no discussion on some recent POMDP-based OPE estimators.
>
> Our primary motivation was to utilize a well-established, standard OPE for offline MDP. This allows us to isolate and analyze the effects of our proposed shadow variable plus the bridge function method for MNAR rewards, without complicating it with additional structural complexities in POMDPs.
>
> We will expand the Related Work section to include a comprehensive discussion on recent OPE estimators for POMDPs (including [Zhang & Jiang, 2025](https://arxiv.org/abs/2503.01134); [Zhang et al., 2024](https://arxiv.org/abs/2402.14703); [Majumdar et al., 2024](https://arxiv.org/pdf/2411.19395), and [Kuang et al., 2025](https://openreview.net/pdf?id=t3LTjkXDJA)). In the Discussion, we will highlight the integration of our MNAR framework with POMDPs.

---

> > ### Author Rebuttal · Reviewer_8YRm · 2026-03-31
> >
> > My questions have been reasonably discussed and answered. The authors have conducted additional experiments on MIMIC-III and pointed out the difference w.r.t. reward shaping. These were my 2 major drawbacks, and hence, I would like to raise my score.

---

> > > ### Author Response · Authors · 2026-04-01
> > >
> > > Thanks a lot for your positive feedback and willingness to increase your rating. We are grateful for your time and efforts to review our paper and give constructive comments!

---

### Official Review · Reviewer_U1Wh · 2026-03-11

**Soundness:** 3
**Presentation:** 3
**Significance:** 3
**Originality:** 3
**Overall Recommendation:** 5
**Confidence:** 3

**Summary:**

This paper studies off-policy evaluation in offline reinforcement learning when rewards in logged data are missing not at random, which introduces selection bias even after conditioning on states and actions. The authors propose an identification strategy using a reward-dependent propensity model with future states as shadow variables and introduce a bridge function to recover the conditional mean reward without explicitly modeling the MNAR mechanism. Based on this identification, they develop a Fitted-Q-Evaluation-style estimator with theoretical guarantees and demonstrate improved performance over existing methods in simulations.

**Compliance With Llm Reviewing Policy:**

Affirmed.

**Final Justification:**

The author have addressed my minor comments, and I am happy to keep my acceptance recommendation.

**Key Questions For Authors:**

See Weaknesses above.

**Limitations:**

See Weaknesses above.

**Strengths And Weaknesses:**

Strengths:

- The problem setting is novel and interesting. The paper formulates missing rewards using an observation indicator that can depend on the state, action, and reward.

- It is a well-written paper with a clear and well-organized structure. The assumptions and data generation process are clearly stated, together with clear descriptions of the method and theoretical results.

- The algorithm and the theoretical results are solid.

Weaknesses:

- It appears that the missing reward setting is related to the semi-supervised reinforcement learning setting. Some related work would be worth mentioning in the Related Work section (e.g., [1, 2]).

- It would be helpful to better demonstrate the advantages of the proposed method by comparing it with:
  - methods that only use the observed rewards,
  - methods that naive use the observed rewards to learn an estimated reward function and then impute the unobserved rewards

Minor comment:

- Line 408: weighting scheme of (Wang et al., 2025) --> weighting scheme of Wang et al., 2025

---

> ### Author Rebuttal · Authors · 2026-03-30
>
> We sincerely thank the reviewer for the positive assessment and encouraging feedback. We address the remaining suggestions below.
>
> > W1. It appears that the missing reward setting is related to the semi-supervised reinforcement learning setting. Some related work would be worth mentioning in the Related Work section.
>
> We agree that our setting is connected to semi-supervised RL, more boardly, the partially labelled sequential decision-making problems. We will expand our Related Work section to discuss semi-supervised RL with rewards only in labeled MDPs ([Finn et al., 2017](https://arxiv.org/pdf/1612.00429)), semi-supervised offline RL with mixed labeled and unlabled trajectories ([Zheng et al., 2023](https://proceedings.mlr.press/v202/zheng23b/zheng23b.pdf)), and offline policy learning for partially reward-labeled trajectories ([Li et al., 2023](https://proceedings.mlr.press/v202/li23b/li23b.pdf)).
>
> In the new version, we will make these connections explicit and clarifying the distinction from out setting, which focus on OPE rather than policy learning or reward modeling.
>
> > W2. It would be helpful to better demonstrate the advantages of the proposed method by comparing it with:
> methods that only use the observed rewards;
> methods that naive use the observed rewards to learn an estimated reward function and then impute the unobserved rewards.
>
> The first is exactly the NaiveFQE baseline in our experiments, which performs FQE using only the observed rewards.
>
> We add a new baseline method ImputeFQE, which first fits a kernel ridge regression on observed $(S_t, A_t, R_t)$ tuples to learn an estimated reward function, then imputes all missing rewards with the predicted values, then performs standard FQE on the completed dataset.
>
> In additional simulations, all methods are evaluated across varying sample sizes and horizon lengths with different missingness rates. As shown in the figures below, NaiveFQE and ImputeFQE both exhibit substantial biases under MNAR that grow with the missing rate, while ProxFQE remains consistently accurate.
>
> The simulation results are visualized in the following figures in anonymized image links
> - https://ibb.co/Hft7XM1M MSE vs sample size under different missingness rates;
> - https://ibb.co/xSF2jjL5 MSE vs horizon length under different missingness rates;
> - https://ibb.co/hFwMwB0t MSE vs reward type under different missingness rates.
>
> Both methods perform worse than ours under MNAR because of the selection bias on observed rewards. NaiveFQE introduces bias by conditioning on $O_t = 1$, which shifts the effective reward distribution. ImputeFQE compounds this issue as its reward model is trained on the same biased observed subset.
>
> We also demonstrate the better performance of our method in analyzing MIMIC-III Sepsis dataset, following the setup of [Raghu et al. (2017)](https://arxiv.org/pdf/1711.09602). The pre-processed dataset has 13,943 ICU patients, 48-dimension states, 25 discrete actions (5-level of Vasopressor dose by 5 level of IV fluid dose). We set horizon length T=10. The reward is set as the negative change in SOFA score $$R_t=-(SOFA_{t+1}-SOFA_t).$$ We apply synthetic MNAR missingness to the observed rewards. The observation indicator $$O_t\sim\text{Bernoulli}(\sigma(c_0+2R_t^{\text{norm}}+0.3S_t^{\text{signal}}-0.1A_t^{\text{norm}})),$$
> where
> - $R_t^{\text{norm}}$ and $A_t^{\text{norm}}$ are normalized rewards and actions,
> - $S_t^{\text{signal}}=\text{normalized Arterial lactate}-0.5*\text{normalized MeanBP}$,
> - The intercept $c_0$ is calibrated to achieve target missing rates of 20%-80%.
>
> To get target policy, we first perform DDQN on the original data. We construct a conservative policy: When $O_{t-1}=0$, both vasopressor and IV fluid dose levels in $A_t$ are reduced by 1 (clipped at 0). This creates action distributions that differ across missingness-aware states. We split the data by patient ID: 60% for training and 40% for evaluating the policy value.
>
> We compare each method's estimated policy value $\hat{V}(\pi)$ against OracleFQE (uses fully observed rewards) and measure the absolute error $|\hat{V}(\pi)-V_{\text{Oracle}}|$ under different missing rates.
>
> The results are visualized in https://ibb.co/P3HWd1c
> - ProxFQE achieves the lowest bias across all missing rates, while all baselines exhibit increasing bias as the MNAR missing rate grows.
> - NaiveFQE and ImputeFQE both exhibit increasing bias as the MNAR missing rate grows, but ImputeFQE is consistently the worst among all methods — roughly 2–3x the bias of NaiveFQE at every rate.
>
> > W3. Line 408: weighting scheme of (Wang et al., 2025) --> weighting scheme of Wang et al., 2025
>
> We fixed this typo in a new version.

---

> > ### Author Rebuttal · Reviewer_U1Wh · 2026-04-01
> >
> > The authors have addressed my minor comments on the manuscript. I therefore maintain my recommendation for acceptance.

---

> > > ### Author Response · Authors · 2026-04-01
> > >
> > > Thanks a lot for your positive feedback. We are grateful for your time and efforts to review our paper and give constructive comments!

---

### Official Review · Reviewer_uu3L · 2026-03-12

**Soundness:** 3
**Presentation:** 3
**Significance:** 3
**Originality:** 3
**Overall Recommendation:** 5
**Confidence:** 2

**Summary:**

There are times in offline Reinforcement Learning where certain data isn't recorded. This work focuses on Missing Not at Random (MNAR) problems, where a reward is not recorded because of the reward itself. To solve this problem the authors utilize the next state ($S_{t+1}$) as a shadow variable to help and impute the missing rewards. The dataset contains not only $(S, A, R, S')$ tuples, but an additional observation variable $O$. Some assumptions around the data is that next state cannot be correlated to the observation variable $O$, however, if the reward is observable, $O=1$, then it should be correlated with the next state. The authors design an unbiased estimator called a bridge function ($b$) that takes in $(s, a, s')$ and acts as a proxy for the mean reward. Training $b$ through standard least squares is biased because of double-sampling, so they instead use a min-max estimator. Once the bridge function is trained, they apply Fitted-Q-Evaluation (FQE) using imputed rewards: $\tilde{R} = O \cdot R + (1-O) \cdot b(S, A, S')$. They addionally provide error bounds for their Prox-FQE algorithm and provide empirical validation of their approach by comparing it against naive FQE and Inverse Probability Weighted FQE to show that their model significantly reduces variance in MNAR settings.

**Compliance With Llm Reviewing Policy:**

Affirmed.

**Final Justification:**

I hold to my original review. My main concern was the motivation of the approach, which was addressed in the authors’ rebuttal. I maintain my positive evaluation, though I have lower confidence in this research area.

**Key Questions For Authors:**

How likely are Assumptions 4.1 and 4.2 to hold in practice? For example, the paper mentions a case where a patient stops visiting the doctor because they have recovered or are seeking emergency care. In such situations, is there even a well-defined next state if the patient never returns? Could the authors provide a concrete real-world example where their algorithm would apply (i.e. where the assumptions would reasonably hold)?

**Limitations:**

Yes

**Strengths And Weaknesses:**

**Soundness**: The paper is thorough, clearly stating its assumptions and providing theorems of existence and uniqueness for cases where the algorithm works. The authors also provide error bounds for their method and prove that it is unbiased. These theoretical results are connected with empirical experiments. The authors also address the fact that reward missingness can still be influenced by unobserved variables.

**Presentation**: I found the paper quite difficult to read, although I suspect that much of this difficulty is due to my unfamiliarity to the material.

**Significance**: From a theoretical perspective, the work appears useful: it provides substantial information about existence, uniqueness, and error bounds. However, I do wonder how often the required assumptions are satisfied in practice.

**Originality**: Much of the work builds on recent ideas (e.g., shadow variables, bridge-based imputation), and MNAR in MDPs has also been discussed previously. However, the authors go into considerable detail to clarify exactly which assumptions are necessary and what results can be proven under those assumptions.

---

> ### Author Rebuttal · Authors · 2026-03-30
>
> We sincerely thank the reviewer for the positive evaluation and for recognizing the thoroughness of our theoretical analysis. We also appreciate the reviewer's honesty regarding the difficulty of the material, and we will work to improve the presentation in the revised version. Below we address the question raised.
>
> > Q1. How likely are Assumptions 4.1 and 4.2 to hold in practice? For example, the paper mentions a case where a patient stops visiting the doctor because they have recovered or are seeking emergency care. In such situations, is there even a well-defined next state if the patient never returns? Could the authors provide a concrete real-world example where their algorithm would apply (i.e. where the assumptions would reasonably hold)?
>
> We agree that Assumptions 4.1, 4.2 are not intended to hold in arbitrary dropout settings.
>
> Our method applies to the setting with **missing reward but observed next-state transitions**, as the identification explicitely uses $S_{t+1}$ as the shadow variable and the formulation assumes the terminal state $S_{T+1}$ is observable. Therefore, if a patient completely leaves the trajectory and no subsequent information is available, then there is no well defined next state for our scheme, instead, that case is better viewed as trajectory truncation or dropout rather than reward missingness setting.
>
> A real-world example is scheduled chronic-disease management with EHR or claims follow-up. At visit $t$, the action is treatment decision, the reward is a patient-reported outcome that may be missing due to the incomplete questionnaire, while $S_{t+1}$ is still observed through later lab tests, prescriptions, encounters, hospitalization records. In this scenario, Assumption 4.2 is plausible since future clinical measurements are informative about the latent health outcome, and Assumption 4.1 is also plausible if the recording decision does not further affect the distribution of subsequent measurements conditioning on $(S_t,A_t,R_t)$. This is the exact scenario targeted by our identification method through the shadow variable.
>
> We will revise the motivating healthcare example accordingly to clearly distinguish between **missing reward with observed follow-up state** and **completely dropout from the system**.
>
> To demonstrate practical usefulness, we add experiments on real-world MIMIC-III sepsis cohort, following [Raghu et al. (2017)](https://arxiv.org/pdf/1711.09602). The preprocessed dataset contains 13,943 ICU patients, 48-dim states, 25 discrete actions (5 Vasopressor levels × 5 IV fluid levels), and horizon T=10. The reward is the decrease in SOFA $$R_t=-(SOFA_{t+1}-SOFA_t).$$ We impose synthetic MNAR missingness on observed rewards. The observation indicator $$O_t\sim Bernoulli(\sigma(c_0+2R_t^{norm}+0.3S_t^{signal}-0.1A_t^{norm})),$$
> where
> - $R_t^{norm}$ and $A_t^{norm}$ are normalized rewards and actions
> - $S_t^{signal}=\text{normalized Arterial lactate}-0.5*\text{normalized MeanBP}$
> - The intercept $c_0$ is calibrated to target missing rates from 0.2 to 0.8.
>
> We first perform DDQN on the original data. We construct a target conservative policy: when $O_{t-1}=0$, both vasopressor and IV fluid dose levels in $A_t$ are reduced by 1 (clipped at 0). This creates action distributions that differ across missingness-aware states. We split the data by patient ID: 60% for training and 40% for evaluating the policy value.
>
> We compare each method with OracleFQE (uses fully observed rewards), and report absolute error $|\hat V(\pi)-V_{Oracle}|$. We also add two new baselines. ImputeFQE fits a regressor on observed rewards using $(S_t,A_t)$, imputes missing rewards, then applies standard FQE. SCOPE will be introduced later.
>
> See results in https://ibb.co/P3HWd1c. ProxFQE achieves the lowest error across all missing rates.

---

> > ### Author Rebuttal · Reviewer_uu3L · 2026-04-03
> >
> > Thank you for the response. All of my questions are answered and I will maintain my positive assessment.

---

> > > ### Author Response · Authors · 2026-04-04
> > >
> > > Thanks a lot for your positive feedback. We are grateful for your time and efforts to review our paper and give constructive comments!

---

### Official Review · Reviewer_CLUo · 2026-03-13

**Soundness:** 4
**Presentation:** 3
**Significance:** 3
**Originality:** 3
**Overall Recommendation:** 4
**Confidence:** 4

**Summary:**

This paper studies the problem of missing-not-at-random (MNAR) rewards in RL, specifically when the reason a reward is missing depends on the reward itself. The authors propose a new approach for OPE using shadow variables constructed from the next state. The next state serves as an extra piece of information that helps uncover the hidden rewards. The authors introduce a set of bridge functions to unbiasedly estimate the reward given a $(\text{state}, \text{action})$ pair. By combining the bridge functions with the shadow variable, they build the identification structure and motivate a doubly robust (DR) estimator. Overall, the paper is interesting in the way it constructs the shadow variable. It is well grounded in OPE and addresses the challenging MNAR problem with solid theoretical analysis.

**Compliance With Llm Reviewing Policy:**

Affirmed.

**Final Justification:**

I will keep my original review and score (weak accept) for this paper.

**Key Questions For Authors:**

See "Strengths And Weaknesses"

**Limitations:**

See "Strengths And Weaknesses"

**Strengths And Weaknesses:**

**Strengths**

1. **Interesting shadow variable construction.** The idea of using the next state as a shadow variable together with bridge functions is interesting. This allows the policy value to be identified without directly modeling the MNAR mechanism, which most existing literature handles through strong assumptions.

2. The theoretical analysis is well presented and comprehensive. The assumptions are clearly stated, and the paper provides error bounds for both bridge estimation and policy value estimation.

3. The empirical results demonstrate the performance of the proposed method compared with existing baselines.

### Weaknesses

1. In Figure 1, the authors show that $O_t$ may causally affect the next-state action $A_{t+1}$ under the target policy, but this arrow does not exist for the behavior policy. I find this confusing. In my understanding, both policies should share the same structural dependencies between variables. The policies may differ in how actions are selected, but the underlying causal relationships should remain the same. In fact, the two policies could even overlap or be identical. In that case, the OPE setting would degenerate, but the structural graph should still be consistent. Could the authors explain why the dependency on $O_t$ differs between the two policies? A concrete example would help clarify this point.

2. For Assumption 3.4, it would be helpful to better understand how restrictive the upper bound on $\kappa_t$ is. Is there an intuitive interpretation of this condition? It would also help if the authors could discuss whether similar assumptions appear in existing literature. More explanation would make this assumption easier to understand.

3. Previous work such as Wang et al. (2025) also uses shadow variables to address MNAR rewards. As discussed in the paper, that approach requires explicitly specifying a valid shadow variable to ensure consistent estimation. In this work, the authors propose using $S_{t+1}$ as the shadow variable and enforce the shadow variable conditions on the next state. From an identification perspective, why is this a more natural choice? It would be helpful if the authors could further clarify why this approach is preferable compared with Wang et al. (2025).

4. The paper would be stronger with additional simulation studies. For example, the authors could vary the missing reward rate (e.g., 0.2, 0.4, 0.6, 0.8) or test different reward generation mechanisms. This would help illustrate the robustness of the proposed method compared with existing approaches.

---

> ### Author Rebuttal · Authors · 2026-03-30
>
> We sincerely thank the reviewer for the careful reading and thoughtful feedback. Below we address each question in detail.
>
> > W1. Could the authors explain why the dependency on $O_t$ differs between the two policies? A concrete example would help clarify this point.
>
> We create this difference to reflect the role of the two policies in the setup. The target policy is allowed to depend on the previous observation indicator $O_{t-1}$ as it is available at decision time $t$ and may influence the next action. The behavior policy is the historical logging policy only depending on the current state. We aim to evaluate a richer class of missingness-aware target policies using data collected under a simpler historical policy.
>
> In healthcare, if the previous lab result $R_{t-1}$ is missing, a target policy at time $t$ may choose a more conservative treatment, whereas the historical standard-care policy may act only on the current clinical state $S_t$ and not explicitly depend on prior missingness. We will modify our introduction to give motivating examples.
>
> > W2. For Assumption 3.4... how restrictive the upper bound on $\kappa_t$ is. Is there an intuitive interpretation... discuss whether similar assumptions appear in existing literature...
>
> Assumption 3.4 is an enhanced version for Assumption 3.3, where $\kappa_t$ measures the worst-case ratio between the target action probability over the behavior action probability at step t. Assumption 3.4 requires that this mismatch is weakening over horizon at a summable rate, i.e., $\sum_t\log\kappa_t<\infty$, so that the cumulative mismatch $\Pi_{j=1}^t \kappa_j$ remains uniformly bounded, which is a technical coverage condition to prevent error amplification from exponentially growing with the horizon.
>
> Coverage/concentrability assumptions are typical in standard offline RL. For example, [Chen & Jiang (2019)](https://arxiv.org/pdf/1905.00360) introduces mild distribution shift requiring sufficient state-action coverage, [Zhan et al.(2022)](https://arxiv.org/pdf/2202.04634) notes that the classical FQI-style OPEs often rely on *strong* all policy concentrability assumptions like our Assumption 3.4. Specifically, similar assumption has been used in Assumption 2 from [Miao et al.(2022)](https://arxiv.org/pdf/2209.10064) in a POMDP setting.
>
> We will include above explanation for Assumption 3.4 in the future version.
>
> > W3. ...authors propose using $S_{t+1}$ as the shadow variable... From an identification perspective, why is this a more natural choice? It would be helpful... compared with Wang et al.(2025).
>
> Our use of $S_{t+1}$ as the shadow variable is natural from the MDP structure and $S_{t+1}$ is already observed in logged transition. From an identification perspective, one of the key advantage is that under Assumption 3.1 (no future dependency) that $O_t\perp(S_{t+1:T}, R_{t+1:T})|(S_t,A_t, R_t)$, Assumption 4.1 automatically holds, so we can use $P(S_{t+1}|R_t,S_t,A_t,O_t=1)=P(S_{t+1}|R_t,S_t,A_t)$ to identify the bridges from the observable subset. Also, the immediate reward is defined accoding to the transition $r_t(s,a,s')=E[R_t|S_t=s,A_t=S_{t+1}=s']$, so $S_{t+1}$ is the most immediate post-action variable that can remain informative about the latent reward.
>
> Compared with Wang et al.(2025), our approach does not require specifying an extra shadow variable $Z_t$, which may depend on expert knowledge. Instead, our $S_{t+1}$ is already in the trajectory.
>
> We will make this comparison more explicit, emphasizing the tradeoff that we rely on relevance and completeness conditions for $S_{t+1}$.
>
> > W4. The paper would be stronger with additional simulation... could vary the missing reward rate... or test different reward generation mechanisms.
>
> We compare 5 methods (ProxFQE, NaiveFQE, IPW-FQE, ImputeFQE, SCOPE) across 50 seeds under missing rate from 0.2 to 0.8, and two reward generation mechanisms. See figures below
> - https://ibb.co/Hft7XM1M MSE vs sample size under different missing rates;
> - https://ibb.co/xSF2jjL5 MSE vs horizon length under different missing rates;
> - https://ibb.co/hFwMwB0t MSE vs reward type under different missing rates.
>
> Specifically
> - Varying missing rate: The missingness indicator $O_t\sim \text{Bernoulli}(\sigma(c_0-0.1A_t+0.2[1,-2]^T S_t+2.5R_t))$, where $c_0$ is calibrated to achieve the target missing rates.
>     - ProxFQE consistently achieves the lowest MSE across all settings, with the advantage growing as missingness increases (https://ibb.co/Hft7XM1M, https://ibb.co/xSF2jjL5).
> - Different reward mechanisms
>     - Sigmoid reward
> $$R_t = \sigma([0.9-0.6A_t,-0.7]^T S_t+[1.3,2]^T S_{t+1}-0.4A_t)+U_t,$$ where $U_t\sim Uniform[-0.1,0.1]$.
>     - Linear reward
> $$R_t=clip([0.5,-0.3]^T S_t+[0.8,0.6]^T S_{t+1}-0.3A_t+\epsilon_t,-1,1),$$ where $\epsilon_t\sim N(0,0.01)$.
>
>     ProxFQE achieves the lowest MSE for both reward types at nearly all missing rates (https://ibb.co/hFwMwB0t).
>
> **We also conduct real data analysis. See other responses.**

---

> > ### Author Rebuttal · Reviewer_CLUo · 2026-04-03
> >
> > I thank the authors for the detailed response, which addressed my concerns. I will maintain my positive evaluation.

---

> > > ### Author Response · Authors · 2026-04-04
> > >
> > > Thanks a lot for your positive feedback. We are grateful for your time and efforts to review our paper and give constructive comments!

---

### Decision · Program_Chairs · 2026-04-30

**Decision:**

Accept (regular)

**Comment:**

This paper studies off-policy evaluation in offline RL with rewards missing not at random, and proposes an identification and estimation framework based on future-state shadow variables, bridge functions, and an FQE-style estimator. Reviewers found the problem important and timely, and viewed the technical development as sound. The main strengths are the novelty of the MNAR reward setting in sequential decision-making, the careful identification strategy, and the accompanying consistency and finite-sample guarantees.

The main concerns were initially about the clarity and practical plausibility of some assumptions, the breadth of related work, and the limited empirical evaluation in the submitted version. In rebuttal, the authors addressed these issues well by clarifying the assumptions and causal interpretation, expanding discussion of related literature, adding stronger baselines, and providing additional synthetic and MIMIC-III experiments. Reviewers who raised these concerns indicated that they were resolved.

Overall, the paper makes a solid technical contribution on an important problem, and the rebuttal satisfactorily addressed the main weaknesses. I recommend accept.